# Dirichlet Energy Constrained Learning for Deep Graph Neural Networks

**Kaixiong Zhou**
Rice University
Kaixiong.Zhou@rice.edu

**Xiao Huang**
The Hong Kong Polytechnic University
xiaohuang@comp.polyu.edu.hk

**Daochen Zha**
Rice University
Daochen.Zha@rice.edu

**Rui Chen**
Samsung Research America
rui.chen1@samsung.com

**Li Li**
Samsung Research America
li.li1@samsung.com

**Soo-Hyun Choi**[*]
Samsung Electronics
soohyunc@gmail.com

**Xia Hu**
Rice University
xia.hu@rice.edu

## Abstract

Graph neural networks (GNNs) integrate deep architectures and topological structure modeling in an effective way. However, the performance of existing GNNs would decrease significantly when they stack many layers, because of the over-smoothing issue. Node embeddings tend to converge to similar vectors when GNNs keep recursively aggregating the representations of neighbors. To enable deep GNNs, several methods have been explored recently. But they are developed from either techniques in convolutional neural networks or heuristic strategies. There is no generalizable and theoretical principle to guide the design of deep GNNs. To this end, we analyze the bottleneck of deep GNNs by leveraging the Dirichlet energy of node embeddings, and propose a generalizable principle to guide the training of deep GNNs. Based on it, a novel deep GNN framework – Energetic Graph Neural Networks (EGNN) is designed. It could provide lower and upper constraints in terms of Dirichlet energy at each layer to avoid over-smoothing. Experimental results demonstrate that EGNN achieves state-of-the-art performance by using deep layers.

## 1 Introduction

Graph neural networks (GNNs) [1] are promising deep learning tools to analyze networked data, such as social networks [2, 3, 4], academic networks [5, 6, 7], and molecular graphs [8, 9, 10, 11]. Based on spatial graph convolutions, GNNs apply a recursive aggregation mechanism to update the representation of each node by incorporating representations of itself and its neighbors [12]. A variety of GNN variations have been explored for different real-world networks and applications [13, 14].

A key limitation of GNNs is that when we stack many layers, the performance would decrease significantly. Experiments show that GNNs often achieve the best performance with less than 3 layers [15, 13]. As the layer number increases, the node representations will converge to indistinguishable vectors due to the recursive neighborhood aggregation and non-linear activation [16, 17].

---

[*]Corresponding author.

35th Conference on Neural Information Processing Systems (NeurIPS 2021).

Such phenomenon is recognized as over-smoothing issue [18, 19, 20, 21, 22]. It prevents the stacking of many layers and modeling the dependencies to high-order neighbors.

A number of algorithms have been proposed to alleviate the over-smoothing issue and construct deep GNNs, including embedding normalization [23, 24, 25], residual connection [26, 27, 28] and random data augmentation [29, 30, 31]. However, some of them are motivated directly by techniques in convolutional neural networks (CNNs) [32], such as the embedding normalization and residual connection. Others are based on heuristic strategies, such as random embedding propagation [30] and dropping edge [29]. Most of them only achieve comparable or even worse performance compared to their shallow models. Recently, a metric of Dirichlet energy has been applied to quantify the over-smoothing [33], which is based on measuring node pair distances. With the increasing of layers, the Dirichlet energy converges to zero since node embeddings become close to each other. But there is a lack of empirical methods to leverage this metric to overcome the over-smoothing issue.

Therefore, it remains a non-trivial task to train a deep GNN architecture due to three challenges. First, the existing efforts are developed from diverse perspectives, without a generalizable principle and analysis. The abundance of these components also makes the design of deep GNNs challenging, i.e., how should we choose a suitable one or combinations for real-world scenarios? Second, even if an effective indicator of over-smoothing is given, it is hard to theoretically analyze the bottleneck and propose a generalizable principle to guide the training of deep GNNs. Third, even if theoretical guidance is given, it may be difficult to be utilized and implemented to train GNNs empirically.

To this end, in this paper, we target to develop a generalizable framework with a theoretical basis, to handle the over-smoothing issue and enable effective deep GNN architectures. In particular, we will investigate two research questions. 1) Is there a theoretical and generalizable principle to guide the architecture design and training of deep GNNs? 2) How can we develop an effective architecture to achieve state-of-the-art performance by stacking a large number of layers? Following these questions, we make three major contributions as follows.

- We propose a generalizable principle – Dirichlet energy constrained learning, to guide the training of deep GNNs by regularizing Dirichlet energy. Without proper training, the Dirichlet energy would be either too small due to the over-smoothing issue, or too large when the node embeddings are over-separating. Our principle carefully defines an appropriate range of Dirichlet energy at each layer. Being regularized within this range, a deep GNN model could be trained by jointly optimizing the task loss and energy value.

- We design a novel deep architecture – Energetic Graph Neural Networks (EGNN). It follows the proposed principle and could efficiently learn an optimal Dirichlet energy. It consists of three components, i.e., orthogonal weight controlling, lower-bounded residual connection, and shifted ReLU (SReLU) activation. The trainable weights at graph convolutional layers are orthogonally initialized as diagonal matrices, whose diagonal values are regularized to meet the upper energy limit and eliminate the over-separating. The residual connection strength is determined by the lower energy limit to avoid the over-smoothing. While the widely-used ReLU activation causes the extra loss of Dirichlet energy, the linear mapping worsens the learning ability of GNNs. We apply SReLU with a trainable shift to provide a trade-off between the non-linear and linear mappings.

- We show that the proposed principle and EGNN can well explain most of the existing techniques for deep GNNs. Empirical results demonstrate that EGNN could be easily trained to reach $64$ layers and achieves surprisingly competitive performance on benchmarks.

## 2   Problem Statement

**Notations.**   Given an undirected graph consisting of $n$ nodes, it is represented as $G = (A, X)$, where $A \in \mathbb{R}^{n \times n}$ denotes the adjacency matrix and $X \in \mathbb{R}^{n \times d}$ denotes the feature matrix. Let $\tilde{A} := A + I_n$ and $\tilde{D} := D + I_n$ be the adjacency and degree matrix of the graph augmented with self-loops. The augmented normalized Laplacian is then given by $\tilde{\Delta} := I_n - \tilde{P}$, where $\tilde{P} := \tilde{D}^{-\frac{1}{2}} \tilde{A} \tilde{D}^{-\frac{1}{2}}$ is an augmented normalized adjacency matrix used for the neighborhood aggregation in GNN models.

**Node classification task.** GNNs have been adopted in many applications [6, 11, 34]. Without loss of generality, we take node classification as an example. Given a graph $G = (A, X)$ and a set of its nodes with labels for training, the goal is to predict the labels of nodes in a test set.

We now use the graph convolutional network (GCN) [15] as a typical example, to illustrate how traditional GNNs perform the network analysis task. Formally, the layer-wise forward-propagation operation in GCN at the $k$-th layer is defined as:

$$X^{(k)} = \sigma(\tilde{P}X^{(k-1)}W^{(k)}). \tag{1}$$

$X^{(k)}$ and $X^{(k-1)}$ are node embedding matrices at layers $k$ and $k-1$, respectively; $W^{(k)} \in \mathbb{R}^{d \times d}$ denotes trainable weights used for feature transformation; $\sigma$ denotes an activation function such as ReLU; $X^{(0)} = X$ at the initial layer of GCN. The embeddings at the final layer are optimized with a node classification loss function, e.g., cross-entropy loss. The recursive neighborhood aggregation in Eq. (1) will make node embeddings similar to each other as the number of layer $k$ increases. This property, i.e., over-smoothing, prevents traditional GNNs from exploring neighbors many hops away. In practice, the dependencies to high-order neighbors are important to the node classification. The traditional shallow GNNs may have sub-optimal performances in the downstream tasks [16, 28].

## 3 Dirichlet Energy Constrained Learning

In this paper, we aim to develop an effective principle to alleviate the over-smoothing issue and enable deep GNNs to leverage the high-order neighbors. We first theoretically analyze the over-smoothing issue, and then provide a principle to explain the key constraint in training deep GNNs.

Node pair distance has been widely adopted to quantify the over-smoothing based on embedding similarities [19, 23]. Among the series of distance metrics, Dirichlet energy is simple and expressive for the over-smoothing analysis [33]. Thus, we adopt Dirichlet energy and formally define it as below.

**Definition 1.** Given node embedding matrix $X^{(k)} = [x_1^{(k)}, \cdots, x_n^{(k)}]^\top \in \mathbb{R}^{n \times d}$ learned from GCN at the $k$-th layer, the Dirichlet energy $E(X^{(k)})$ is defined as follows:

$$E(X^{(k)}) = \text{tr}(X^{(k)^\top} \tilde{\Delta} X^{(k)}) = \frac{1}{2} \sum a_{ij} || \frac{x_i^{(k)}}{\sqrt{1+d_i}} - \frac{x_j^{(k)}}{\sqrt{1+d_j}} ||_2^2, \tag{2}$$

where $\text{tr}(\cdot)$ denotes trace of a matrix; $a_{ij}$ is edge weight given by the $(i, j)$-th element in matrix $A$; $d_i$ is node degree given by the $i$-th diagonal element in matrix $D$. Dirichlet energy reveals the embedding smoothness with the weighted node pair distance. While a smaller value of $E(X^{(k)})$ is highly related to the over-smoothing, a larger one indicates that the node embeddings are over-separating even for those nodes with the same label. Considering the node classification task, one would prefer to have an appropriate Dirichlet energy at each layer to separate the nodes of different classes while keeping those of the same class close. However, under some conditions, the upper bound of Dirichlet energy is theoretically proved to converge to 0 in the limit of infinite layers [33]. In other words, all nodes converge to a trivial fixed point in the embedding space.

Based on the previous analysis, we derive the corresponding lower bound and revisit the over-smoothing/separating problem from the model design and training perspectives. To simplify the derivation process, we remove the non-linear activation $\sigma$, and re-express GCN as: $X^{(k)} = P \cdots PXW^{(1)} \cdots W^{(k)}$. The impact of non-linear function will be considered in the model design.

**Lemma 1.** The Dirichlet energy at the $k$-th layer is bounded as follows:

$$(1-\lambda_1)^2 s_{\min}^{(k)} E(X^{(k-1)}) \leq E(X^{(k)}) \leq (1-\lambda_0)^2 s_{\max}^{(k)} E(X^{(k-1)}). \tag{3}$$

The detailed proof is provided in the Appendix. $\lambda_1$ and $\lambda_0$ are the non-zero eigenvalues of matrix $\tilde{\Delta}$ that are most close to values 1 and 0, respectively. $s_{\min}^{(k)}$ and $s_{\max}^{(k)}$ are the squares of minimum and maximum singular values of weight $W^{(k)}$, respectively. Note that the eigenvalues of $\tilde{\Delta}$ vary with the real-world graphs, and locate within range $[0, 2)$. We relax the above bounds as below.

**Lemma 2.** The lower and upper bounds of Dirichlet energy at the $k$-th layer could be relaxed as:

$$0 \leq E(X^{(k)}) \leq s_{\max}^{(k)} E(X^{(k-1)}). \tag{4}$$

Besides the uncontrollable eigenvalues determined by the underlying graph, it is shown that the Dirichlet energy can be either too small or too large without proper design and training on weight $W^{(k)}$. On one hand, based on the common Glorot initialization [35] and L2 regularization, we empirically find that some of the weight matrices approximate to zero in a deep GCN. The corresponding square singular values are hence close to zero in these intermediate layers. That means the Dirichlet energy will become zero at the higher layers of GCN and causes the over-smoothing issue. On the other hand, without the proper weight initialization and regularization, a large $s_{\max}^{(k)}$ may lead to the energy explosion and the over-separating.

The Dirichlet energy plays a key role in training a deep GNN model. However, the optimal value of Dirichlet energy varies in the different layers and applications. It is hard to be specified ahead and then enforces the node representation learning. Therefore, we propose a principle – Dirichlet energy constrained learning, defined in Proposition 1. It provides appropriate lower and upper limits of Dirichlet energy. Regularized by such a given range, a deep GNN model could be trained by jointly optimizing the node classification loss and Dirichlet energy at each layer.

**Proposition 1.** Dirichlet energy constrained learning defines the lower & upper limits at layer $k$ as:

$$c_{\min} E(X^{(k-1)}) \leq E(X^{(k)}) \leq c_{\max} E(X^{(0)}). \tag{5}$$

We apply the transformed initial feature through trainable function $f: X^{(0)} = f(X) \in \mathbb{R}^{n \times d}$. Both $c_{\min}$ and $c_{\max}$ are positive hyperparameters. From value interval $(0, 1)$, hyperparameter $c_{\min}$ is selected by satisfying constraint of $E(X^{(k)}) \geq c_{\min}^k E(X^{(0)}) > 0$. In such a way, the over-smoothing is overcome since the Dirichlet energies of all the layers are larger than appropriate limits related to $c_{\min}^k$. Compared with the initial transformed feature $X^{(0)}$, the intermediate node embeddings of the same class are expected to be merged closely to have a smaller Dirichlet energy and facilitate the downstream applications. Therefore, we exploit the upper limit $c_{\max} E(X^{(0)})$ to avoid over-separating, where $c_{\max}$ is usually selected from $(0, 1]$. In the experiment part, we show that the optimal energy accompanied with the minimized classification loss locates within the above range at each layer. Furthermore, hyperparameters $c_{\min}$ and $c_{\max}$ could be easily selected from the large and appropriate value scopes, which do not affect the model performance.

Given both the low and upper limits, an intuitive solution to search the optimal energy is to train GNNs by optimizing the following constrained problem:

$$
\begin{aligned}
\min \quad & \mathcal{L}_{\text{task}} + \gamma \sum_k \|W^{(k)}\|_F, \\
\text{s.t.} \quad & c_{\min} E(X^{(k-1)}) \leq E(X^{(k)}) \leq c_{\max} E(X^{(0)}), \text{for } k = 1, \cdots, K.
\end{aligned} \tag{6}
$$

$\mathcal{L}_{\text{task}}$ denotes the cross-entropy loss of node classification task; $K$ is layer number of GNN; $\|\cdot\|_F$ denotes Frobenius norm of a matrix; and $\gamma$ is loss hyperparameter. Note that Dirichlet energy has also been adopted to regularize the node representation learning in shallow neural networks [36, 37, 38]. We instead focus on optimizing deep GNNs as shown in Eq. (6), where $K$ is often large.

# 4 Energetic Graph Neural Networks - EGNN

It is non-trivial to optimize Problem (6) due to the expensive computation of $E(X^{(k)})$. Furthermore, the numerous constraints make the problem a very complex optimization hyper-planes, at which the raw task objective tends to fall into local optimums. Instead of directly optimizing Problem (6), we propose an efficient model EGNN to satisfy the constrained learning from three perspectives: weight controlling, residual connection and activation function. We introduce them one by one as follows.

## 4.1 Orthogonal Weight Controlling

According to Lemma 2, without regularizing the maximum square singular value $s_{\max}^{(k)}$ of matrix $W^{(k)}$, the upper bound of Dirichlet energy can be larger than the upper limit, i.e., $s_{\max}^{(k)} E(X^{(k-1)}) > c_{\max} E(X^{(0)})$. That means the Dirichlet energy of a layer may break the upper limit of constrained learning, and makes Problem (6) infeasible. In this section, we show how to satisfy such limit by controlling the singular values during weight initialization and model regularization.

**Orthogonal initialization.** Since the widely-used initialization methods (e.g., Glorot initialization) fail to restrict the scopes of singular values, we adopt the orthogonal approach that initializes trainable weight $W^{(k)}$ as a diagonal matrix with explicit singular values [39]. To restrict $s_{\max}^{(k)}$ and meet the constrained learning, we apply an equality constraint of $s_{\max}^{(k)}E(X^{(k-1)}) = c_{\max}E(X^{(0)})$ at each layer. Based on this condition, we derive Proposition 2 to initialize those weights $W^{(k)}$ and their square singular values for all the layers of EGNN, and give Lemma 3 to show how we can satisfy the upper limit of constrained learning. The detailed derivation and proof are listed in Appendix.

**Proposition 2.** At the first layer, weight $W^{(1)}$ is initialized as a diagonal matrix $\sqrt{c_{\max}} \cdot I_d$, where $I_d$ is identity matrix with dimension $d$ and the square singular values are $c_{\max}$. At the higher layer $k > 1$, weight $W^{(k)}$ is initialized with an identity matrix $I_d$, where the square singular values are 1.

**Lemma 3.** Based on the above orthogonal initialization, at the starting point of training, the Dirichlet energy of EGNN satisfies the upper limit at each layer $k$: $E(X^{(k)}) \leq c_{\max}E(X^{(0)})$.

**Orthogonal regularization.** However, without proper regularization, the initialized weights cannot guarantee they will still satisfy the constrained learning during model training. Therefore, we propose a training loss that penalizes the distances between the trainable weights and initialized weights $\sqrt{c_{\max}}I_d$ or $I_d$. To be specific, we modify the optimization problem (6) as follows:

$$\min \mathcal{L}_{\text{task}} + \gamma||W^{(1)} - \sqrt{c_{\max}}I_d||_F + \gamma \sum_{k=2}^{K} ||W^{(k)} - I_d||_F. \tag{7}$$

Comparing with the original problem (6), we instead use the weight penalization to meet the upper limit of constrained learning, and make the model training efficient. While a larger $\gamma$ highly regularizes the trainable weights around the initialized ones to satisfy the constrained learning, a smaller $\gamma$ assigns the model more freedom to adapt to task data and optimize the node classification loss. Considering the above orthogonal initialization where weight $W^{(k)}$ is diagonal and sparse, we use the simplest distance constraint in Eq. (7) to update weight at the vicinity of its initialization. The singular values of updated sparse weight will be mainly determined by the dominant diagonal values, which are potentially close to the initialized ones. Therefore, we are able to control the singular values and regularize the upper limit of Dirichlet energy even at the model training phase. In the future work, more the advanced orthogonal initialization and regularization approaches could be explored to further boost performance of deep GNNs [40, 41, 42].

## 4.2 Lower-bounded Residual Connection

Although the square singular values are initialized and regularized properly, we may still fail to guarantee the lower limit of constrained learning in some specific graphs. According to Lemma 1, the lower bound of Dirichlet energy is $(1 - \lambda_1)^2 s_{\min}^{(k)}E(X^{(k-1)})$. In the real-world applications, eigenvalue $\lambda_1$ may exactly equal to 1 and relaxes the lower bound as zero as shown in Lemma 2. For example, in Erdős–Rényi graph with dense connections [43], the eigenvalues of matrix $\tilde{\Delta}$ converge to 1 with high probability [17]. Even though $s_{\min}^{(k)} > 0$, the Dirichlet energy can be smaller than the lower limit and leads to the over-smoothing. To tackle this problem, we adopt residual connections to the initial layer $X^{(0)}$ and the previous layer $X^{(k-1)}$. To be specific, we define the residual graph convolutions as:

$$X^{(k)} = \sigma([(1 - c_{\min})\tilde{P}X^{(k-1)} + \alpha X^{(k-1)} + \beta X^{(0)}]W^{(k)}). \tag{8}$$

$\alpha$ and $\beta$ are residual connection strengths determined by the lower limit of constrained learning, i.e., $\alpha + \beta = c_{\min}$. We are aware that the residual technique has been used before to set up deep GNNs [26, 44, 28]. However, they either apply the whole residual components, or combine an arbitrary fraction without theoretical insight. Instead, we use an appropriate residual connection according to the lower limit of Dirichlet energy. In the experiment part, we show that while a strong residual connection overwhelms information in the higher layers and reduces the classification performance, a weak one will lead to the over-smoothing. In the following, we justify that both the lower and upper limits in the constrained learning can be satisfied with the proposed lower-bounded residual connection. The detailed proofs are provided in Appendix.

**Lemma 4.** Suppose that $c_{\max} \geq c_{\min}/(2c_{\min} - 1)^2$. Based upon the orthogonal controlling and residual connection, the Dirichlet energy of initialized EGNN is larger than the lower limit at each layer $k$, i.e., $E(X^{(k)}) \geq c_{\min}E(X^{(k-1)})$.

**Lemma 5.** Suppose that $\sqrt{c_{\max}} \geq \frac{\beta}{(1-c_{\min})\lambda_0+\beta}$. Being augmented with the orthogonal controlling and residual connection, the Dirichlet energy of initialized EGNN is smaller than the upper limit at each layer $k$, i.e., $E(X^{(k)}) \leq c_{\max}E(X^{(0)})$.

### 4.3 SReLU Activation

Note that the previous theoretical analysis and model design are conducted by ignoring the activation function, which is usually given by ReLU in GNN. In this section, we first theoretically discuss the impact of ReLU on the Dirichlet energy, and then demonstrate the appropriate choice of activation.

**Lemma 6.** We have $E(\sigma(X^{(k)})) \leq E(X^{(k)})$ if activation function $\sigma$ is ReLU or Leaky-ReLU [33].

It is shown that the application of ReLU further reduces the Dirichlet energy, since the negative embeddings are non-linearly mapped to zero. Although the trainable weights and residual connections are properly designed, the declining Dirichlet energy may violate the lower limit. On the other hand, a simplified GNN with linear identity activation will have limited model learning ability although it does not change the energy value. For example, simple graph convolution (SGC) model achieves comparable performance with the traditional GCN only with careful hyperparameter tuning [45]. We propose to apply SReLU to achieve a good trade-off between the non-linear and linear activations [46, 47]. SReLU is defined element-wisely as:

$$\sigma(X^{(k)}) = \max(b, X^{(k)}), \tag{9}$$

where $b$ is a trainable shift shared for each feature dimension of $X^{(k)}$. SReLU interpolates between the non-linearity and linearity depending on shift $b$. While the linear identity activation is approximated if $b$ is close to $\infty$, the non-linear mapping is activated if node embedding is smaller than the specific $b$. In our experiments, we initialize $b$ with a negative value to provide an initial trade-off, and adapt it to the given task by back-propagating the training loss.

### 4.4 Connections to Previous Work

Recently, various techniques have been explored to enable deep GNNs [16, 24, 30]. Some of them are designed heuristically from diverse perspectives, and others are analogous to CNN components without theoretical insight tailored to graph analytics. In the following, we show how our principle and EGNN explain the existing algorithms, and expect to provide reliable theoretical guidance to the future design of deep GNNs.

**Embedding normalization.** The general normalization layers, such as pair [23], batch [25] and group [24] normalizations, have been used to set up deep GNNs. The pair normalization (PairNorm) aims to keep the node pair distances as a constant in the different layers, and hence relieves the over-smoothing. Motivated from CNNs, the batch and group normalizations re-scale the node embeddings of a batch and a group, respectively. Similar to the operation in PairNorm, they learn to maintain the node pair distance in the node batch or group. The adopted Dirichlet energy is also a variant of the node pair distance. The existing normalization methods can be regarded as training GNN model with a constant energy constraint. However, this will prevent GNN from optimizing the energy as analyzed in Section 3. We instead regularize it within the lower and upper energy limits, and let model discover the optimum.

**Dropping edge.** As a data augmentation method, dropping edge (DropEdge) randomly masks a fraction of edges at each epoch [29]. It makes graph connections sparse and relieves the over-smoothing by reducing information propagation. Specially, the contribution of DropEdge could be explained from the perspective of Dirichlet energy. In Erdős–Rényi graph, eigenvalue $\lambda_0$ converges to 1 if the graph connections are more and more dense [17]. DropEdge reduces the value of $\lambda_0$, and helps improve the upper bound of Dirichlet energy $(1 - \lambda_0)^2 s_{\max}^{(k)} E(X^{(k-1)})$ to slow down the energy decreasing speed. In the extreme case where all the edges are dropped in any a graph,

Laplacian $\tilde{\Delta}$ becomes a zero matrix. As a result, we have eigenvalue $\lambda_0$ of zero and maximize the upper bound. In practice, the dropping rate has to be determined carefully depending on various tasks. Instead, our principle assigns model freedom to optimize the Dirichlet energy within a large and appropriate range.

**Residual connection.** Motivated from CNNs, residual connection has been applied to preserve the previous node embeddings and relieve the over-smoothing. Especially, the embedding from the last layer is reused and combined completely in related work [26, 48, 49]. A fraction of the initial embedding is preserved in model GCNII [28] and APPNP [50]. Networks JKNet [27] and DAGNN [51] aggregate all the previous embeddings at the final layers. The existing work uses the residua connection empirically. In this work, we derive and explain the residual connection to guarantee the lower limit of Dirichlet energy. By modifying hyperparameter $c_{\min}$, our EGNN can easily evolve to the existing deep residual GNNs, such as GCNII and APPNP.

**Model simplification.** Model SGC [45] removes all the activation and trainable weights to avoid over-fitting issue, and simplifies the training of deep GNNs. It is equivalent to EGNN with $c_{max} = 1$ and $b = -\infty$, where weights $W^{(k)}$ and shifts $b$ are remained as constants. Such simplification will reduce the model learning ability. As shown in Eq. (7), we adopt loss hyperparameter $\gamma$ to learn the trade-off between maintaining the orthogonal weights or updating them to model data characteristics.

# 5 Experiments

In this section, we empirically evaluate the effectiveness of EGNN on real-world datasets. We aim to answer the following questions. **Q1:** How does our EGNN compare with the state-of-the-art deep GNN models? **Q2:** Whether or not the Dirichlet energy at each layer of EGNN satisfies the constrained learning? **Q3:** How does each component of EGNN affect the model performance? **Q4:** How do the model hyperparameters impact the performance of EGNN?

## 5.1 Experiment Setup

**Datasets.** Following the practice of previous work, we evaluate EGNN by performing node classification on four benchmark datasets: Cora, Pubmed [52], Coauthor-Physics [53] and Ogbn-arxiv [54]. The detailed statistics are listed in Appendix.

**Baselines.** We consider seven state-of-the-art baselines: GCN [15], PairNorm [23], DropEdge [29], SGC [45], JKNet [27], APPNP [50], and GCNII [28]. They are implemented based on their open repositories. The detailed descriptions of these baselines are provided in Appendix.

**Implementation.** We implement all the baselines using Pytorch Geometric [55] based on their official implementations. The model hyperparameters are reused according to the public papers or are fine-tuned by ourselves if the classification accuracy could be further improved. Specially, we apply max-pooling to obtain the final node representation at the last layer of JKNet. In Ogbn-arxiv, we additionally include batch normalization between the successive layers in all the considered GNN models except PairNorm. Although more tricks (e.g., label reusing and linear transformation as listed in leader board) could be applied to improve node classification in Ogbn-arxiv, we focus on comparing the original GNN models in enabling deep layer stacking. The training hyperparameters are carefully set by following the previous common setting and are listed in Appendix.

We implement our EGNN upon GCN, except for the components of weight initialization and regularization, lower-bounded residual connection and SReLU. We choose hyperparameters $c_{\max}$, $c_{\min}$, $\gamma$ and $b$ based on the validation set. For the weight initialization, we set $c_{\max}$ to be 1 for all the datasets; that is, the trainable weights are initialized as identity matrices at all the graph convolutional layers. The loss hyperparameter $\gamma$ is 20 in Cora, Pubmed and Coauthor-Physics to strictly regularize towards the orthogonal matrix; and it is $10^{-4}$ in Ogbn-arxiv to improve the model's learning ability. For the lower-bounded residual connection, we choose residual strength $c_{\min}$ from range $[0.1, 0.75]$ and list the details in Appendix. The trainable shift $b$ is initialized with $-10$ in Cora and Pubmed; it is initialized to $-5$ and $-1$ in Coauthor-Physics and Ogbn-arxiv, respectively. We also study these hyperparameters in the following experiments. All the experiment results are the averages of 10 runs.

Table 1: Node classification accuracies in percentage with various depths: 2, 16, 32/64. The highest accuracy at each column is in bold.

| Datasets | Cora | | | Pubmed | | | Coauthors-Physics | | | Ogbn-arxiv | | |
|---|---|---|---|---|---|---|---|---|---|---|---|---|
| Layer Num | 2 | 16 | 64 | 2 | 16 | 64 | 2 | 16 | 32 | 2 | 16 | 32 |
| GCN | 82.5 | 22.0 | 21.9 | **79.7** | 37.9 | 38.4 | 92.4 | 13.5 | 13.1 | 70.4 | 70.6 | 68.5 |
| PairNorm | 74.5 | 44.2 | 14.2 | 73.8 | 68.6 | 60.0 | 86.3 | 84.0 | 83.6 | 67.6 | 70.4 | 69.6 |
| DropEdge | 82.7 | 23.6 | 25.2 | 79.6 | 45.9 | 40.0 | 92.5 | 85.1 | 35.2 | 70.5 | 70.4 | 67.1 |
| SGC | 75.7 | 72.1 | 24.1 | 76.1 | 70.2 | 38.2 | 92.2 | 91.7 | 84.8 | 69.2 | 64.0 | 59.5 |
| JKNet | 80.8 | 74.5 | 70.0 | 77.2 | 70.0 | 66.1 | **92.7** | 92.2 | 91.6 | **70.6** | 71.8 | 71.4 |
| APPNP | 82.9 | 79.4 | 79.5 | 79.3 | 77.1 | 76.8 | 92.3 | 92.7 | 92.6 | 68.3 | 65.5 | 60.7 |
| GCNII | 82.4 | 84.6 | 85.4 | 77.5 | 79.8 | 79.9 | 92.5 | 92.9 | 92.9 | 70.1 | 71.5 | 70.5 |
| EGNN | **83.2** | **85.4** | **85.7** | 79.2 | **80.0** | **80.1** | 92.6 | **93.1** | **93.3** | 68.4 | **72.7** | **72.7** |

## 5.2 Experiment Results

**Node classification results.** To answer research question **Q1**, Table 1 summarizes the test classification accuracies. Each accuracy is averaged over 10 random trials. We report the results with 2/16/64 layers for Cora and Pubmed, and 2/16/32 layers for Coauthor-Physics and Ogbn-arxiv. Due to space limit, we report the detailed results of mean accuracy and standard deviation in Appendix.

We observe that our EGNN generally outperforms all the baselines across the four datasets, especially in the deep cases ($K \geq 16$). Notably, the node classification accuracy is consistently improved with the layer stacking in EGNN until $K = 32$ or $64$, which demonstrates the benefits of deep graph neural architecture to leverage neighbors multiple hops away. While the state-of-the-art models PairNorm, DropEdge, SGC, JKNet, and APPNP alleviate the over-smoothing issue to some extend, their performances still drop with the increasing of layers. Most of their 32/64-layer models are even worse than their corresponding shallow versions. As the most competitive deep architecture in literature, GCNII augments the transformation matrix as $(1 - \phi)I_d + \phi W^{(k)}$, where $0 < \phi < 1$ is a hyperparameter to preserve the identity mapping and enhance the minimum singular value of the augmented weight. Instead of explicitly defining the strength of identity mapping, we propose the orthogonal weight initialization based on the upper limit of Dirichlet energy and apply the orthogonal weight regularization. Based on Eq. (7), EGNN automatically learns the optimal trade-off between identity mapping and task adaption. Furthermore, we use SReLU activation and the residual connection to theoretically control the lower limit of Dirichlet energy. The experimental results show that EGNN not only outperforms GCNII in the small graphs Cora, Pubmed and Coauthor-Physics, but also delivers significantly superior performance in the large graph Obgn-arxiv, achieving 3.1% improvement over GCNII with 32 layers.

**Dirichelet energy visualization.** To answer research question **Q2**, we show the Dirichlet energy at each layer of a 64-layer EGNN in Cora and Pubmed datasets in Figure 1. To have better visualization purposes, by keeping other default hyperparameters unchanged, EGNN is trained with $c_{max}/c_{min} = 0.4/0.15$ and $c_{max}/c_{min} = 0.4/0.11$ in Cora and Pubmed, respectively. We only plot and

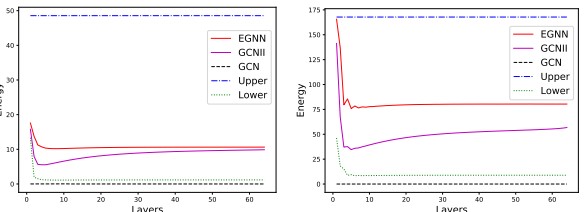

Figure 1: Dirichelet energy variation with layers in Cora (**Left**) and Pubmed (**Right**). The upper and lower denotes the energy limits.

compare with the baseline approaches of GCN and GCNII due to space limit. For other methods, the Dirichlet energy is either close to zero or overly large due to the over-smoothing issue or over-separating issue of node embeddings, respectively.

It is shown that the Dirichlet energies of EGNN are strictly constrained within the range determined by the lower and upper limits of the constrained learning. Due to the over-smoothing issue in GCN, all the node embeddings converge to zero vectors. GCNII has comparable or smaller Dirichlet energy by carefully and explicitly designing both the initial connection and identity mapping strengths. In contrast, our EGNN only gives the appropriate limits of Dirichlet energy, and let the model learn the

Table 2: Ablation studies on weight initialization, lower limit $c_{\min}$ and activation function of EGNN.

| Component | Type | Cora | | | Pubmed | | | Coauthors-Physics | | | Ogbn-arxiv | | |
|---|---|---|---|---|---|---|---|---|---|---|---|---|---|
| | | 2 | 16 | 64 | 2 | 16 | 64 | 2 | 16 | 32 | 2 | 16 | 32 |
| Weight | Glorot | 77.8 | 40.2 | 23.6 | 68.4 | 62.6 | 60.2 | 92.6 | 81.7 | 73.4 | 68.4 | **72.8** | 72.7 |
| initialization | Orthogonal | 83.2 | 85.4 | **85.7** | **79.2** | **80.0** | **80.1** | 92.6 | **93.1** | **93.3** | 68.4 | 72.7 | **72.7** |
| Lower | 0. | **83.6** | 68.6 | 12.9 | 78.9 | 77.1 | 44.1 | **92.8** | 91.4 | 79.7 | **70.9** | 69.4 | 62.4 |
| limit setting | $0.1 \sim 0.75$ | 83.2 | 85.4 | **85.7** | **79.2** | **80.0** | **80.1** | 92.6 | **93.1** | **93.3** | 68.4 | 72.7 | **72.7** |
| $c_{\min}$ | 0.95 | 65.4 | 72.0 | 71.5 | 74.0 | 75.3 | 75.7 | 89.4 | 90.4 | 90.5 | 56.5 | 66.8 | 69.5 |
| | Linear | 83.1 | **85.6** | 85.5 | 79.2 | 79.9 | 79.9 | 92.6 | 93.1 | 93.1 | 64.8 | 72.5 | 71.0 |
| Activation | SReLU | 83.2 | 85.4 | **85.7** | **79.2** | **80.0** | **80.1** | 92.6 | **93.1** | **93.3** | 68.4 | 72.7 | **72.7** |
| | ReLU | 83.1 | 85.2 | 85.0 | 79.1 | 79.7 | 79.9 | 92.6 | 93.1 | 93.1 | 68.6 | 72.4 | 72.4 |

optimal energy at each layer for a specific task. The following hyperparameter studies will show that the values of $c_{\min}$ and $c_{\max}$ could be easily selected from a large appropriate range.

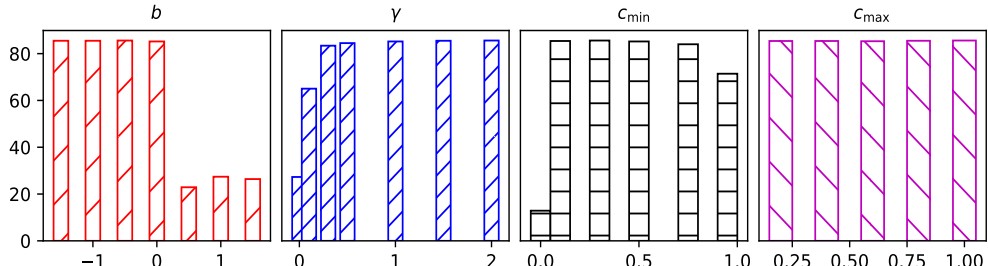

Figure 2: The impacts of hyperparameters $b$, $\gamma$, $c_{\min}$ and $c_{\max}$ on 64-layer EGNN trained in Cora. Y-axis is test accuracy in percent.

**Ablation studies of EGNN components.** To demonstrate how each component affects the training of graph neural architecture and answer research question **Q3**, we perform the ablation experiments with EGNN on all the datasets. For the component of orthogonal weight initialization and regularization, we compare and replace them with the traditional Glorot initialization and Frobenius norm regularization as shown in Eq. (6). Considering the component of lower-bounded residual connection, we vary the lower limit hyperparameter $c_{\min}$ from 0, $0.1 \sim 0.75$ and 0.95. Within the range of $0.1 \sim 0.75$, the adoption of specific values is specified for each dataset in Appendix. The component of the activation function is studied from candidates of linear identity activation, SReLU, and ReLU. Table 2 reports the results of the above ablation studies.

The orthogonal weight initialization and regularization are crucial to train the deep graph neural architecture. In Cora, Pubmed, and Coauthor-Physics, Glorot initialization and Frobenius norm regularization fail to control the singular values of trainable weights, which may lead to overly large or small Dirichlet energy and affect the node classification performance. In Ogbn-arxiv, the input node features are described by dense word embeddings of a paper [56], where the trainable weights in GNN are required to capture data statistics and optimize the classification task. EGNN applies a small loss hyperparameter $\gamma$ of $10^{-4}$ to let the model adapt to the given task, which is equivalent to the traditional regularization. Therefore, the two approaches have comparable performances.

An appropriate lower limit could enable the deep EGNN. While the Dirichlet energy may approach zero without the residual connection, the overwhelming residual information with $c_{\min} = 0.95$ prevents the higher layer from learning the new neighborhood information. Within the large and appropriate range of $[0.1, 0.75]$, $c_{\min}$ could be easily selected to achieve superior performance.

Activation SReLU performs slightly better than the linear identity activation and ReLU. This is because SReLU could automatically learn the trade-off between linear and non-linear activations, which prevents the significant dropping of Dirichlet energy and ensures the model learning ability.

**Hyperparameter analysis.** To understand the hyperparameter impacts on a 64-layer EGNN and answer research question **Q4**, we conduct experiments with different values of initial shift $b$, loss

factor $\gamma$, lower limit factor $c_{\min}$ and upper one $c_{\max}$. We present the hyperparameter study in Figure 2 for Cora, and show the others with similar tendencies in Appendix.

We observe that our method is not sensitive to the choices of $b$, $\gamma$, $c_{\min}$ and $c_{\max}$ in a wide range: (i) The initial shift value should be $b \leq 0$, in order to avoid the overly nonlinear mapping and Dirichlet energy damage. (ii) It is shown that EGNN approximates the optimal performance once the loss factor $\gamma$ is larger than a specific threshold. The thresholds are $0.3$ in Cora, $0.1$ in Pubmed and Coauthor-Physics, and 1es-4 in Ogbn-arxiv, respectively. The threshold depends on the specific dataset: while a larger potentially works in the small dataset to strictly regularize Dirichlet energy, a smaller one would be preferred for the large dataset to capture the complex data manifold. (iii) $c_{\min}$ within the appropriate range $[0.1, 0.75]$ allows the model to expand neighborhood size and preserve residual information to avoid the over-smoothing. (iv) As shown in Figure 1, since energy $E(X^{(k)})$ at the hidden layer is much smaller than $E(X^{(0)})$ from the input layer, we could easily satisfy the upper limit with $c_{\max}$ in a large range $[0.2, 1]$. Given these large hyperparameter ranges, EGNN could be easily trained with deep layers.

## 6    Conclusions

In this paper, we propose a Dirichlet energy constrained learning principle to show the importance of regularizing the Dirichlet energy at each layer within reasonable lower and upper limits. Such energy constraint is theoretically proved to help avoid the over-smoothing and over-separating issues. We then design EGNN based on our theoretical results and empirically demonstrate that the constrained learning plays a key role in guiding the design and training of deep graph neural architecture. The detailed analysis is presented to illustrate how our principle connects and combines the previous deep methods. The experiments on benchmarks show that EGNN could be easily trained to achieve superior node classification performances with deep layer stacking. We believe that the constrained learning principle will help discover deeper and more powerful GNNs in the future.

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
