# A Appendix

## A.1 Dataset Statistics

We conduct experiments on four benchmark graph datasets, including Cora, Pubmed, Coauthor-Physics and Ogbn-arxiv. They are widely used to study the over-smoothing issue and test the performance of deep GNNs. We use the public train/validation/test split in Cora and Pubmed, and randomly split Coauthor-Physics by following the previous practice. Their data statistics are summarized in Table 3.

Table 3: Data statistics.

| Datasets | # Nodes | # Edges | # Classes | # Features | # Train/Validation/Test nodes | Setting |
|---|---|---|---|---|---|---|
| Cora | 2,708 | 5,429 | 7 | 1,433 | 140/500/1,000 | Transductive (one graph) |
| Pubmed | 19,717 | 44,338 | 3 | 500 | 60/500/1,000 | Transductive (one graph) |
| Coauthor-Physics | 34,493 | 247,962 | 5 | 8,415 | 100/150/34,243 | Transductive (one graph) |
| Ogbn-arxiv | 169,343 | 1,166,243 | 40 | 128 | 90,941/29,799/48,603 | Transductive (one graph) |

## A.2 Baselines

To validate the effectiveness of the Dirichlet energy constrained learning principle and our EGNN on the node classification problem, we consider baseline GCN and other state-of-the-art deep GNNs based upon GCN. They are summarized as follows:

- **GCN [15].** It is mathematically defined in Eq. (1), which learns the node embeddind by simply propagating messages over the normalized adjacency matrix.

- **PairNorm [23].** Based upon GCN, PairNorm is applied between the successive graph convolutional layers to normalize node embeddings and to alleviate the over-smoothing issue.

- **DropEdge [29].** It randomly removes a certain number of edges from the input graph at each training epoch, which reduces the convergence speed of over-smoothing.

- **SGC [45].** It simplies the vanilla GCN by removing all the hidden weights and activation functions, which could avoid the over-fitting issue in GCN.

- **Jumping knowledge network (JKNet) [27].** Based upon GCN, all the hidden node embeddings are combined at the last layer to adapt the effective neighborhood size for each node. Herein we apply max-pooling to combine the the series of node embeddings from the hidden layers.

- **Approximate personalized propagation of neural predictions (APPNP) [50].** It applies personalized PageRank to improve the message propagation scheme in vanilla GCN. Furthermore, APPNP simplifies model by removing the hidden weight and activation function and preserving a small fraction of initial embedding at each layer.

- **Graph convolutional network via initial residual and identity mapping (GCNII) [28].** It is an extension of the vanilla GCN model with two simple techniques at each layer: an initial connection to the input feature and an identity mapping added to the trainable weight.

## A.3 Implementation Details

For each experiment, we train with a maximum of 1500 epochs using the Adam optimizer and early stopping. Following the previous common settings in the considered benchmarks, we list the key training hyperparameters for each of them in Table 4. All the experiment results are reported by the averages of 10 independent runs.

## A.4 Lower Limit Setting

We carefully choose the lower limit hyperparameter $c_{\min}$ from range $[0.1, 0.75]$ for each dataset based on the classification performance and Dirichlet energy on the validation set. Note that we have the residual connection strengths $\alpha$ and $\beta$ which satisfy constraint: $\alpha + \beta = c_{\min}$. Specially, we use $c_{\min}$ of 0.2 (layer number $K < 32$) and 0.15 ($K \geq 32$) in Cora, where $\alpha, \beta = 0.1$ in all the layer cases. We use $c_{\min}$ of 0.12 ($K < 32$) and 0.11 ($K \geq 32$) in Pubmed, where $\beta = c_{\min}$ and $\alpha = 0$.

Table 4: The training hyperparameter settings in benchmarks.

| Dataset | Dropout rate | Weight decay (L2) | Learning rate | # Training epoch |
|---|---|---|---|---|
| Cora | 0.6 | $5 \cdot 10^{-4}$ | $5 \cdot 10^{-3}$ | 1500 |
| Pubmed | 0.5 | $5 \cdot 10^{-4}$ | $1 \cdot 10^{-2}$ | 1500 |
| Coauthor-Physics | 0.6 | $5 \cdot 10^{-5}$ | $5 \cdot 10^{-3}$ | 1500 |
| Ogbn-arxiv | 0.1 | 0 | $3 \cdot 10^{-3}$ | 1000 |

We apply $c_{\min}$ of 0.12 in Coauthor-Physics, where $\beta = 0.1$ and $\alpha = 0.02$. We apply $c_{\min}$ and $\beta$ of 0.6 and 0.1 when $K < 32$, respectively; we make use of $c_{\min}$ and $\beta$ of 0.75 and 0.25 if $K \geq 32$, respectively. In these two cases, the residual connection strength $\alpha$ to the last layer is 0.5.

### A.5 Proof for Lemma 1

**Lemma 1.** The Dirichlet energy at the $k$-th layer is bounded as follows:

$$(1 - \lambda_1)^2 s_{\min}^{(k)} E(X^{(k-1)}) \leq E(X^{(k)}) \leq (1 - \lambda_0)^2 s_{\max}^{(k)} E(X^{(k-1)}).$$

**Proof.** By ignoring the activation function, we obtain the upper bound as below.

$$
\begin{aligned}
E(X^{(k)}) &= E(\tilde{P} X^{(k-1)} W^{(k)}) \\
&= \operatorname{tr}((\tilde{P} X^{(k-1)} W^{(k)})^\top \tilde{\Delta} (\tilde{P} X^{(k-1)} W^{(k)})) \\
&= \operatorname{tr}((\tilde{P} X^{(k-1)})^\top \tilde{\Delta} (\tilde{P} X^{(k-1)}) W^{(k)} (W^{(k)})^\top) \\
&\leq \operatorname{tr}((\tilde{P} X^{(k-1)})^\top \tilde{\Delta} (\tilde{P} X^{(k-1)})) \sigma_{\max}(W^{(k)}(W^{(k)})^\top) \\
&= \operatorname{tr}((\tilde{P} X^{(k-1)})^\top \tilde{\Delta} (\tilde{P} X^{(k-1)})) s_{\max}^{(k)} \\
&= \operatorname{tr}([(I_n - \tilde{\Delta}) X^{(k-1)}]^\top \tilde{\Delta} [(I_n - \tilde{\Delta}) X^{(k-1)}]) s_{\max}^{(k)} \\
&= \operatorname{tr}((X^{(k-1)})^\top \tilde{\Delta} (I_n - \tilde{\Delta})^2 X^{(k-1)}) s_{\max}^{(k)} \\
&\leq (1 - \lambda_0)^2 \operatorname{tr}((X^{(k-1)})^\top \tilde{\Delta} X^{(k-1)}) s_{\max}^{(k)} \\
&= (1 - \lambda_0)^2 s_{\max}^{(k)} E(X^{(k-1)}).
\end{aligned}
$$

$\sigma_{\max}(\cdot)$ denotes the maximum eigenvalue of a matrix, and $\tilde{P} = I_n - \tilde{\Delta}$. Since $\operatorname{tr}(X^\top \tilde{\Delta} X) \geq 0$, where $X \in \mathbb{R}^{n \times d}$ is a feature matrix, we can obtain the inequality relationship: $\operatorname{tr}(X^\top \tilde{\Delta} X W^{(k)} (W^{(k)})^\top) \leq \operatorname{tr}(X^\top \tilde{\Delta} X) \sigma_{\max}(W^{(k)} (W^{(k)})^\top)$. In a similar way, we can also get the upper bound of $(1 - \lambda_0)^2 s_{\max}^{(k)} E(X^{(k-1)})$.

Similarly, we derive the lower bound as below.

$$
\begin{aligned}
E(X^{(k)}) &= E(\tilde{P} X^{(k-1)} W^{(k)}) \\
&= \operatorname{tr}((\tilde{P} X^{(k-1)} W^{(k)})^\top \tilde{\Delta} (\tilde{P} X^{(k-1)} W^{(k)})) \\
&\geq \operatorname{tr}((\tilde{P} X^{(k-1)})^\top \tilde{\Delta} (\tilde{P} X^{(k-1)})) \sigma_{\min}(W^{(k)}(W^{(k)})^\top) \\
&= \operatorname{tr}((\tilde{P} X^{(k-1)})^\top \tilde{\Delta} (\tilde{P} X^{(k-1)})) s_{\min}^{(k)} \\
&= \operatorname{tr}((X^{(k-1)})^\top \tilde{\Delta} (I_n - \tilde{\Delta})^2 X^{(k-1)}) s_{\min}^{(k)} \\
&\geq (1 - \lambda_1)^2 \operatorname{tr}((X^{(k-1)})^\top \tilde{\Delta} X^{(k-1)}) s_{\min}^{(k)} \\
&= (1 - \lambda_1)^2 s_{\min}^{(k)} E(X^{(k-1)}).
\end{aligned}
$$

### A.6 Derivation of Proposition 2

All the trainable weights at the graph convolutional layers of EGNN are initialized as the orthogonal diagonal matrices. At the first layer, the upper bound of Dirichlet energy is given by $s_{\max}^{(1)} E(X^{(0)})$. Given constraint $s_{\max}^{(1)} E(X^{(0)}) = c_{\max} E(X^{(0)})$, we can obtain $W^{(1)} = \sqrt{c_{\max}} \cdot I_d$. The square singular values are then restricted as: $s_{\min}^{(1)} = s_{\max}^{(1)} = c_{\max}$. For layer $k > 1$, we further relax the upper bound as: $s_{\max}^{(k)} E(X^{(k-1)}) \leq \prod_{j=1}^{k} s_{\max}^{(j)} E(X^{(0)})$. Note that $s_{\max}^{(1)} = c_{\max}$ at the first layer. Given constraint $\prod_{j=1}^{k} s_{\max}^{(j)} E(X^{(0)}) = c_{\max} E(X^{(0)})$, we can obtain weight $W^{(k)} = I_d$. The square singular values are restricted as: $s_{\min}^{(k)} = s_{\max}^{(k)} = 1$, and $\prod_{j=1}^{k} s_{\max}^{(j)} = c_{\max}$.

## A.7 Proof for Lemma 3

**Lemma 3.** Based on the above orthogonal initialization, at the starting point of training, the Dirichlet energy of EGNN satisfies the upper limit at each layer $k$: $E(X^{(k)}) \leq c_{\max} E(X^{(0)})$.

**Proof.** According to Lemma 2, the Dirichlet energy at layer $k$ is limited as:

$$
\begin{aligned}
E(X^{(k)}) \quad &\leq s_{\max}^{(k)} E(X^{(k-1)}) \\
&\leq \textstyle\prod_{j=1}^{k} s_{\max}^{(j)} E(X^{(0)}) \\
&= c_{\max} E(X^{(0)}).
\end{aligned}
$$

## A.8 Proof for Lemma 4

**Lemma 4.** Suppose that $c_{\max} \geq c_{\min}/(2c_{\min} - 1)^2$. Based upon the orthogonal controlling and residual connection, the Dirichlet energy of initialized EGNN is larger than the lower limit at each layer $k$, i.e., $E(X^{(k)}) \geq c_{\min} E(X^{(k-1)})$.

**Proof.** To obtain the Dirichlet energy relationship between $E(X^{(k)})$ and $E(X^{(k-1)})$, we first expand node embedding $X^{(k)}$ as the series summation in terms of the initial node embedding $X^{(0)}$. We then re-express the graph convolution at layer $k$, which is simplified to depend only on node embedding $X^{(k-1)}$. As a result, we can easily derive Lemma 4. The detailed proofs are provided in the following.

According to Eq. (8), by ignoring the activation function $\sigma$, we obtain the residual graph convolution at layer $k$ as:

$$
\begin{aligned}
X^{(k)} \quad &= [(1 - c_{\min})\tilde{P}X^{(k-1)} + \alpha X^{(k-1)} + \beta X^{(0)}]W^{(k)}. \\
&= [(1 - c_{\min})\tilde{P} + \alpha I_n]X^{(k-1)}W^{(k)} + \beta X^{(0)}W^{(k)},
\end{aligned}
$$

where $I_n$ is an identity matrix with dimension $n$, and $\alpha + \beta = c_{\min}$. We define $Q \triangleq (1 - c_{\min})\tilde{P} + \alpha I_n$, and then simply the above graph convolution as:

$$
X^{(k)} = QX^{(k-1)}W^{(k)} + \beta X^{(0)}W^{(k)}. \tag{10}
$$

To facilitate the proof, we further expand the above graph convolution as the series summation in terms of the initial node embedding $X^{(0)}$ as:

$$
X^{(k)} = Q^k X^{(0)} \prod_{j=1}^{k} W^{(j)} + \beta \sum_{i=0}^{k-1} (Q^i X^{(0)} \prod_{j=k-i}^{k} W^{(j)}),
$$

where the weight matrix product is defined as: $\prod_{j=1}^{k} W^{(j)} \triangleq W^{(1)}W^{(2)} \cdots W^{(k)}$, and $Q^0 \triangleq I_n$. Notably, in our EGNN, the trainable weight $W^{(1)}$ at the first layer is orthogonally initialized as diagonal matrix of $\sqrt{c_{\max}} \cdot I_d$, while $W^{(j)}$ at layer $j > 1$ is initialized as identity matrix $I_d$. Therefore, the series expansion of $X^{(k)}$ could be simplified as:

$$
\begin{aligned}
X^{(k)} \quad &= (\sqrt{c_{\max}}Q^k + \beta\sqrt{c_{\max}}Q^{k-1} + \beta \sum_{i=0}^{k-2} Q^i)X^{(0)} \\
&\triangleq Z^{(k)} X^{(0)}.
\end{aligned}
$$

$Z^{(1)} \triangleq \sqrt{c_{\max}}Q + \beta\sqrt{c_{\max}}I_n$ at the case $k = 1$. Note that $Z^{(k)}$ is invertible if all the eigenvalues of matrix $Q$ are not equal to zero, which could be achieved by selecting an appropriate $\alpha$ depending on the downstream task. Let $\tilde{Z}^{(k)} = [Z^{(k)}]^{-1}$. We then represent the initial node embedding $X^{(0)}$ as: $X^{(0)} = \tilde{Z}^{(k)} X^{(k)}$. Similarly, $X^{(0)} = \tilde{Z}^{(k-1)} X^{(k-1)}$ at layer $k - 1$. Therefore, we can re-express the graph convolution at layer $k$ in Eq. (10) as:

$$
X^{(k)} = (Q + \beta\tilde{Z}^{(k-1)})X^{(k-1)}W^{(k)}
$$

According to Lemma 1, the lower bound of Dirichlet energy at layer $k$ is given by:

$$
E(X^{(k)}) \quad \geq \lambda_{\min}^2(Q + \beta\tilde{Z}^{(k-1)})s_{\min}^{(k)} E(X^{(k-1)}).
$$

$\lambda_{\min}^2(\cdot)$ denotes the minimum square eigenvalue of a matrix. To get the minimum square eigenvalue, we represent the eigenvalue decomposition of matrix $Q$ as: $Q = V\Lambda V^{-1}$, where $V \in \mathbb{R}^{n \times n}$ is the eigenvector matrix and $\Lambda \in \mathbb{R}^{n \times n}$ is the diagonal eigenvalue matrix. We then decompose $(Q + \beta\tilde{Z}^{(k-1)})$ as:

$$\begin{aligned} Q + \beta\tilde{Z}^{(k-1)} \quad &= Q + \beta(\sqrt{c_{\max}}Q^{k-1} + \beta\sqrt{c_{\max}}Q^{k-2} + \beta\sum_{i=0}^{k-3}Q^i)^{-1} \\ &= V\Lambda V^{-1} + \beta V(\sqrt{c_{\max}}\Lambda^{k-1} + \beta\sqrt{c_{\max}}\Lambda^{k-2} + \beta\sum_{i=0}^{k-3}\Lambda^i)^{-1}V^{-1}. \end{aligned}$$

Let $\lambda_Q$ denote the eigenvalue of matrix $Q$. Recalling $Q \triangleq (1 - c_{\min})\tilde{P} + \alpha I_n$ and $\tilde{P} \triangleq I_n - \tilde{\Delta}$. Since the eigenvalues of $\tilde{P}$ are within $(-1, 1]$, we have $-(1 - c_{\min}) + \alpha < \lambda_Q \leq (1 - c_{\min}) + \alpha < 1$. To ensure that $Q$ is invertible, we could apply a larger value of $\alpha$ to have $-(1 - c_{\min}) + \alpha > 0$. The square eigenvalue of matrix $(Q + \beta\tilde{Z}^{(k-1)})$ is:

$$\lambda^2(Q + \beta\tilde{Z}^{(k-1)}) = (\lambda_Q + \beta(\sqrt{c_{max}}\lambda_Q^{k-1} + \beta\sqrt{c_{max}}\lambda_Q^{k-2} + \beta\frac{1 - \lambda_Q^{k-2}}{1 - \lambda_Q})^{-1})^2$$

It could be easily validated that $\frac{\partial \log(\lambda^2(Q+\beta\tilde{Z}^{(k-1)}))}{\partial k} \geq 0$. That means the square eigenvalue increases with the layer $k$. Considering the extreme case of $k \to \infty$, we obtain $\lambda^2(Q + \beta\tilde{Z}^{(k-1)}) \to 1$. Since $s_{\min}^{(k)} = 1$ at layer $k > 1$, we thus obtain $E(X^{(k)}) \geq E(X^{(k-1)}) \geq c_{\min}E(X^{(k-1)})$ when $k \to \infty$. In practice, since $\lambda^2(Q + \beta\tilde{Z}^{(k-1)})$ approximates to one with the increasing of layer $k$, the Dirichlet energy will be maintained as a constant at the higher layers of EGNN, which is empirically validated in Figure 1.

The minimum square eigenvalue is achieved when $k = 1$, i.e., $\lambda_{\min}^2(Q + \beta\tilde{Z}^{(0)})$, where $\tilde{Z}^{(0)} = I_n$ and $\lambda_Q$ is close to $-(1 - c_{\min}) + \alpha$. In this case, we obtain $\lambda_{\min}^2(Q + \beta\tilde{Z}^{(0)}) = (2c_{\min} - 1)^2$. At layer $k = 1$, we have $s_{\min}^{(1)} = c_{\max}$. Since $E(X^{(1)}) \geq \lambda_{\min}^2(Q + \beta\tilde{Z}^{(0)})c_{\max}E(X^{(0)})$, to make sure $E(X^{(1)}) \geq c_{\min}E(X^{(0)})$ at the first layer, we only need to satisfy the following condition:

$$\begin{aligned} &\lambda_{\min}^2(Q + \beta\tilde{Z}^{(0)})c_{\max} \geq c_{\min} \\ \Rightarrow \quad &c_{\max} \geq c_{\min}/(2c_{\min} - 1)^2. \end{aligned}$$

Note that the square eigenvalue is increasing with $k$, and $s_{\min}^{(k)} = 1 \geq c_{\max}$ for layer $k > 1$. At the higher layer $k > 1$, we have $\lambda_{\min}^2(Q + \beta\tilde{Z}^{(k-1)})s_{\min}^{(k)} \geq \lambda_{\min}^2(Q + \beta\tilde{Z}^{(0)})c_{\max}$. Therefore, once the condition of $c_{\max} \geq c_{\min}/(2c_{\min} - 1)^2$ is satisfied, we can obtain $\lambda_{\min}^2(Q + \beta\tilde{Z}^{(k-1)})s_{\min}^{(k)} \geq c_{\min}$ and $E(X^{(k)}) \geq c_{\min}E(X^{(k-1)})$ for all the layers $k$ in EGNN.

### A.9 Proof for Lemma 5

**Lemma 5.** Suppose that $\sqrt{c_{\max}} \geq \frac{\beta}{(1-c_{\min})\lambda_0+\beta}$. Being augmented with the orthogonal controlling and residual connection, the Dirichlet energy of initialized EGNN is smaller than the upper limit at each layer $k$, i.e., $E(X^{(k)}) \leq c_{\max}E(X^{(0)})$.

**Proof.** According to the proof of Lemma 4, we have $X^{(k)} = Z^{(k)}X^{(0)}$. Based on Lemma 1, the upper bound of Dirichlet energy at layer $k$ is given by:

$$E(X^{(k)}) \leq \lambda_{\max}^2(Z^{(k)})E(X^{(0)}),$$

where $\lambda_{\max}^2(\cdot)$ is the maximum square eigenvalue of a matrix. According to the definition of $Z^{(k)}$ and the eigenvalue decomposition of $Q$ in the proof of Lemma 4, we decompose $Z^{(k)}$ as:

$$\begin{aligned} Z^{(k)} \quad &= (\sqrt{c_{\max}}Q^k + \beta\sqrt{c_{\max}}Q^{k-1} + \beta\sum_{i=0}^{k-2}Q^i) \\ &= V(\sqrt{c_{\max}}\Lambda^k + \beta\sqrt{c_{\max}}\Lambda^{k-1} + \beta\sum_{i=0}^{k-2}\Lambda^i)V^{-1}. \end{aligned}$$

Therefore, the square eigenvalue of $Z^{(k)}$ is given by:

$$\lambda^2(Z^{(k)}) = (\sqrt{c_{max}}\lambda_Q^k + \beta\sqrt{c_{max}}\lambda_Q^{k-1} + \beta\frac{1 - \lambda_Q^{k-1}}{1 - \lambda_Q})^2,$$

Table 5: The mean node classification accuracies and standard deviations in percentage on Cora and Pubmed with various depths: 2, 16, 64. The highest accuracy at each column is in bold.

| Datasets | Cora | | | Pubmed | | |
|---|---|---|---|---|---|---|
| Layer Num | 2 | 16 | 64 | 2 | 16 | 64 |
| GCN | 82.52±0.45 | 22.02±6.24 | 21.86±8.04 | **79.66±0.29** | 37.94±0.53 | 38.42±1.01 |
| PairNorm | 74.46±3.13 | 44.23±7.26 | 14.22±1.93 | 73.84±0.90 | 68.59±7.30 | 60.03±10.23 |
| DropEdge | 82.73±0.60 | 23.64±7.61 | 25.18±9.20 | 79.55±0.50 | 45.92±7.14 | 39.97±1.14 |
| SGC | 75.68±0.04 | 72.10±0.00 | 24.10±0.00 | 76.05±0.05 | 70.20±0.00 | 38.19±0.03 |
| JKNet | 80.84±0.66 | 74.54±3.72 | 70.01±7.66 | 77.15±0.68 | 69.98±6.26 | 66.16±8.35 |
| APPNP | 82.94±0.56 | 79.38±0.62 | 79.49±0.66 | 79.33±0.48 | 77.07±0.66 | 76.83±0.90 |
| GCNII | 82.42±0.45 | 84.55±0.43 | 85.44±0.58 | 77.49±1.91 | 79.83±0.56 | 79.94±0.33 |
| EGNN | **83.18±0.24** | **85.36±0.35** | **85.71±0.55** | 79.17±0.34 | **79.99±0.36** | **80.10±0.26** |

where $\lambda_Q$ denotes the eigenvalue of matrix $Q$. Recalling $Q \triangleq (1 - c_{\min})\tilde{P} + \alpha I_n$ and $\tilde{P} \triangleq I_n - \tilde{\Delta}$. The maximum square eigenvalue $\lambda_{\max}^2(Z^{(k)})$ is achieved when $\lambda_Q$ takes the largest value, i.e., $\lambda_Q = \theta_0 = (1 - c_{\min})(1 - \lambda_0) + \alpha$, where $\lambda_0$ is the non-zero eigenvalue of matrix $\tilde{\Delta}$ that is most close to value 0. Therefore, we have $\lambda_{\max}^2(Z^{(k)}) = c_{\max}(\theta_0^k + \beta\theta_0^{k-1} + \frac{\beta(1-\theta_0^{k-1})}{\sqrt{c_{\max}}(1-\theta_0)})^2$. To ensure that $E(X^{(k)}) \le c_{\max}E(X^{(0)})$ for all the layers, we have to satisfy the condition of $\lambda_{\max}^2(Z^{(k)}) \le c_{\max}$. Since $\theta_0 > 0$, we simplify this condition in the followings:

$$\lambda_{\max}^2(Z^{(k)}) \le c_{\max}$$
$$\Rightarrow \quad \theta_0^k + \beta\theta_0^{k-1} + \frac{\beta(1-\theta_0^{k-1})}{\sqrt{c_{\max}}(1-\theta_0)} \le 1$$
$$\Rightarrow \quad \frac{\beta(1-\theta_0^{k-1})}{(1-\theta_0^k - \beta\theta_0^{k-1})(1-\theta_0)} \le \sqrt{c_{\max}}$$
$$\Rightarrow \quad \frac{\beta(1-\theta_0^{k-1})}{(1-\theta_0^{k-1}(\beta+\theta_0))(1-\theta_0)} \le \sqrt{c_{\max}}$$
$$\Rightarrow \quad \frac{\beta(1-\theta_0^{k-1})}{(1-\theta_0^{k-1}(1-(1-c_{\min})\lambda_0))(1-\theta_0)} \le \sqrt{c_{\max}}$$

Note that $0 < (1 - (1 - c_{\min})\lambda_0) < 1$ and $1 - \theta_0^{k-1} < 1 - \theta_0^{k-1}(1 - (1 - c_{\min})\lambda_0)$. The above condition can be satisfied if $\frac{\beta}{1-\theta_0} \le \sqrt{c_{\max}}$. Note that $1 - \theta_0 = (1 - c_{\min})\lambda_0 + \beta$. Therefore, if $\sqrt{c_{\max}} \ge \frac{\beta}{(1-c_{\min})\lambda_0+\beta}$, we obtain $E(X^{(k)}) \le c_{\max}E(X^{(0)})$ for all the layers in EGNN. Such condition can be easily satisfied by adopting $c_{\max} = 1$.

## A.10 Proof for Lemma 6

**Lemma 6.** We have $E(\sigma(X^{(k)})) \le E(X^{(k)})$ if activation function $\sigma$ is ReLU or Leaky-ReLU [33].

**Proof.** Herein we directly adopt the proof from [33] to support the self-containing in this paper. Let $c_1, c_2 \in \mathbb{R}_+$, and let $a, b \in \mathbb{R}$. We have the following relationships:

$$|c_1 a - c_2 b| \quad \ge |\sigma(c_1 a) - \sigma(c_2 b)| \\ = |c_1\sigma(a) - c_2\sigma(b)|.$$

The first inequality holds for activation function $\sigma$ whose Lipschitz constant is smaller than 1, including ReLU and Leaky-ReLU. The second equality holds because $\sigma(cx) = c\sigma(x), \forall c \in \mathbb{R}_+$ and $x \in \mathbb{R}$. Recalling the Dirichlet energy definition in Eq. (2): $E(X^{(k)}) = \frac{1}{2}\sum a_{ij}||\frac{x_i^{(k)}}{\sqrt{1+d_i}} - \frac{x_j^{(k)}}{\sqrt{1+d_j}}||_2^2$. By extending to the vector space and replacing $c_1, c_2, a$, and $b$ with $\frac{1}{\sqrt{1+d_i}}, \frac{1}{\sqrt{1+d_j}}, x_i^{(k)}$, and $x_j^{(k)}$, respectively, we can obtain $E(\sigma(X^{(k)})) \le E(X^{(k)})$.

## A.11 Node Classification Results

Comparing with the baseline approaches of deep GNNs, we list their mean accuracies and standard deviations in Table 5 and 6.

Table 6: The mean node classification accuracies and standard deviations in percentage on Coauthors-Physics and Ogbn-arxiv with various depths: 2, 16, 32. The highest accuracy at each column is in bold.

| Datasets | Coauthors-Physics | | | Ogbn-arxiv | | |
|---|---|---|---|---|---|---|
| Layer Num | 2 | 16 | 32 | 2 | 16 | 32 |
| GCN | 92.36±0.14 | 13.55±2.88 | 13.13±2.80 | 70.42±0.21 | 70.64±0.25 | 68.51±0.96 |
| PairNorm | 86.32±1.06 | 84.02±3.34 | 83.56±5.08 | 67.56±0.14 | 70.37±0.15 | 69.63±0.21 |
| DropEdge | 92.46±0.14 | 85.10±3.02 | 35.23±17.5 | 70.49±0.23 | 70.41±0.55 | 67.14±1.78 |
| SGC | 92.19±0.00 | 91.74±0.00 | 84.80±0.01 | 69.19±0.04 | 64.01±0.05 | 59.46±0.07 |
| JKNet | **92.71±0.19** | 92.15±0.49 | 91.65±1.35 | **70.62±0.11** | 71.85±0.15 | 71.44±0.35 |
| APPNP | 92.33±0.15 | 92.65±0.46 | 92.63±0.42 | 68.26±0.78 | 65.47±0.23 | 60.71±0.16 |
| GCNII | 92.49±0.36 | 92.87±0.23 | 92.94±0.15 | 70.09±0.27 | 71.46±0.16 | 70.52±0.30 |
| EGNN | 92.59±0.09 | **93.10±0.16** | **93.31±0.12** | 68.41±0.25 | **72.74±0.23** | **72.74±0.35** |

## A.12 Hyperparameter Analysis

To further understand the hyperparameter impacts on EGNN and answer research question **Q4**, we conduct more experiments and show in Figures 3, 4 and 5 for Pubmed, Coauthor-Physics and Ogbn-arxiv, respectively.

Similar to the hyperparameter study on Cora, we observe that our method is consistently not sensitive to the choices of $b$, $\gamma$, $c_{\min}$ and $c_{\max}$ within wide value ranges for all the datasets. The appropriate ranges of $b$, $\gamma$, $c_{\min}$ and $c_{\max}$ are $(-\infty, 0]$, $[1, \infty]$, $[0.1, 0.75]$ and $[0.2, 1]$, respectively. Specially, in the large graph of Ogbn-arxiv, our model could even has a large initialization range for $b$. Given these wide hyperparameter ranges, EGNN could be easily constructed and trained with deep layers.

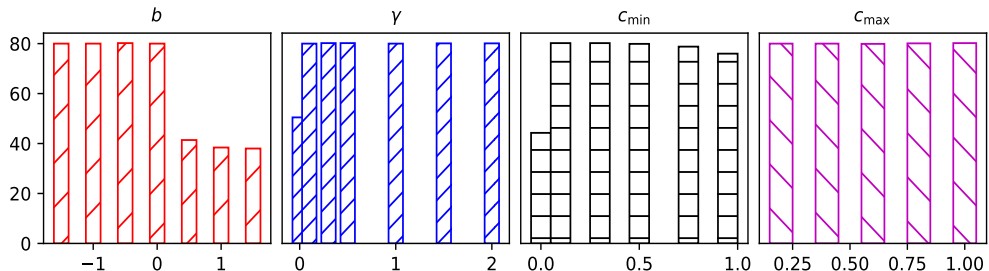

Figure 3: The impacts of hyperparameters $b$, $\gamma$, $c_{\min}$ and $c_{\max}$ on $64$-layer EGNN trained in Pubmed. Y-axis is test accuracy in percent.

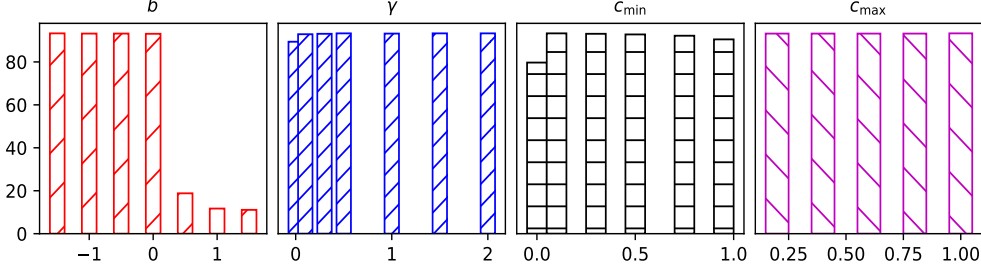

Figure 4: The impacts of hyperparameters $b$, $\gamma$, $c_{\min}$ and $c_{\max}$ on $32$-layer EGNN trained in Coauthor-Physics. Y-axis is test accuracy in percent.

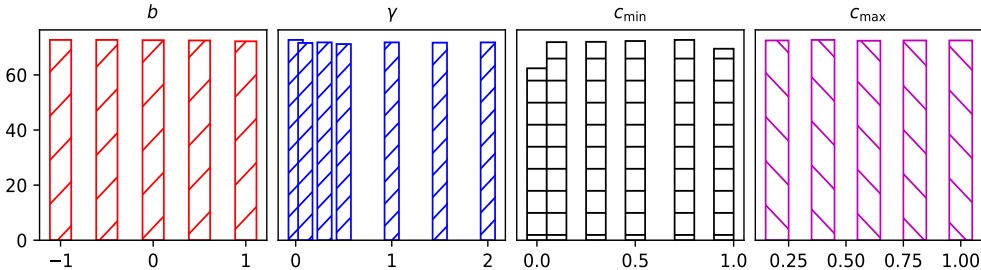

Figure 5: The impacts of hyperparameters $b$, $\gamma$, $c_{\min}$ and $c_{\max}$ on 32-layer EGNN trained in Ogbn-arxiv. Y-axis is test accuracy in percent.