# OpenReview forum: "Dirichlet Energy Constrained Learning for Deep Graph Neural Networks"
_NeurIPS.cc/2021/Conference — NeurIPS 2021 Poster_

### Official Review · Reviewer_TzoX · 2021-07-15

**Rating:** 6
**Confidence:** 4

**Summary:**

The authors suggest a way to prevent over smoothing in many graph neural networks that use the graph Laplacian as the spatial convolution kernel. The idea is to keep the Dirichlet Energy under control throughout the network, and hence not allow the network to oversmooth.

**Limitations And Societal Impact:**

No mentioning.

**Main Review:**

The paper is interesting with several theoretical results that look correct as far as I can tell. The results are nice but a bit limited to four data sets only of the same type, and the method still relies on the x^0 input term of GCNII at every layer, which looks specific to this type of data sets. More specific details are given below:

1) "Orthogonal initialization": the authors apply a diagonal initialization, probably with diagonal entries of +-1. Is that right? The writing is a bit confusing. What happens when the matrices W are not square?

2) Eq. (7) the authors regularize W to be a scaled identity. I would not necessarily call it "orthogonal". In orthogonal regularization, I would expect something like ||W^T*W - c*I||. Why call it or even regularize it this way? Am I missing something?

3) In continuation to the previous remark, the authors say that for three of the four data sets (e.g. cora, pubmed and Coauthors-Physics), they use a high regularization parameter gamma. It means that many W's are close to the identity. That also means that the network computes a lot of FLOPs but the weights W are not so meaningful. Is that a good network design? What happens if there are no W's at all (just W=c*I)?

4) In continuation to the previous points, you penalize the energy, not necessarily constrain it. For a low gamma, the energy can be high. Am I missing something? In your fourth experiment, you use a very small gamma. Is the energy bounded there?

5) Page 9: overwhelming residual information. This is quite obvious. By setting cmin=0.95 you more or less ignore the Laplacian and weight, and just propagate the input through the network without doing anything. I'm not so sure this is even worthy of discussing.

6) In figure 1, it looks like the legend is incorrect. "GCN" and "Lower" are flipped. Am I wrong?

7) You use gamma=20 for the three data sets and the gamma=10^-4 for the last one. That is a huge difference. Can you comment on why is that?

8) "We implement our EGNN upon GCN". The additions in (8) look very similar to those applied in GCNII, so it looks like most of the performance boost is coming from there. Now, the beta term in GCNII (having the input x^0 in all layers) is a bit specific to these data sets (i.e. not used in other scenarios like in standard CNNs or other GNN tasks). How does the method perform without beta? (which is learnable, right?)  Also - in Fig 1 it looks like GCNII does not lose energy because of such additions - so why do we still need to penalize the energy?

9) Are the results for the semi-supervised or fully supervised node classification. I guess the former, but this is not written in the paper.


Post rebuttal:
==========
I upgrade my score to 6.



**Time Spent Reviewing:**

5

---

> ### Author Response · Authors · 2021-08-10
> **Response to Reviewer TzoX**
>
> We really appreciate the reviewer for the agreements on the interestingness and the theoretical contributions in this work. We address the concerns one by one in the following.
>
> Q1 and Q2: Orthogonal initialization and constraint.
>
> The scaled identity weight initialization is a special
> case of orthogonal initialization. Despite the existence of other orthogonal initialization methods, we adopt the scaled identity initialization due to its simplicity and expressiveness to define the eigenvalues along the diagonal. For the non-square weight $W\in \mathbb{R}^{d_1\times d_2}$, it  could still be initialized along the main diagonal only if $d_1\leq d_2$ and $WW^{\top}$ is not singular. In our simi-supervised node classification problem, we empirically observe the scaled identity initialization and the accompanied distance constraint perform even better than other orthogonal initialization and constraining methods, such as Delta orthogonal initialization. One of the possible reason is that weight $W$ will be sparse based on our diagonal initialization and regularization. The eigenvalues of $W$ are mainly determined by the dominant diagonal values, which are potentially remained the same as the initialized ones. Therefore, the Dirichlet energy will be well constrained to achieve the superior performance. In the revision, we will add more discussion on the identity weight and other existing orthogonal weight initialization.
>
> The key contribution in the orthogonal weight controlling of EGNN is that we point out the importance of weight initialization and regularization in training deep GNNs, which is ignored in the previous works. Comparing with the traditional Glorot initialization, we show how to limit the upper energy through carefully initializing the eigenvalues of $W$, and provide theoretical proof in Lemma 3 to demonstrate our idea. The ablation study in Table 2 further empirically validates the crucial role of orthogonal weight controlling. In the future development, more the advanced orthogonal initialization and regularization methods could be proposed to improve deep GNNs.
>
>
> Q3: What happens if there are no W's at all (just W=c*I)?
>
> Actually, SGC is a simplified model by setting all weights $W$ to be identity matrices, which results in much poor performances than those of EGNN. To further clarify the confusion in terms of orthogonal weight controlling, we provide supplementary experiments by comparing EGNN and its variant with constant matrices $W=cI$ on all the four concerned datasets. Given the orthogonal diagonal initialization in EGNN, the following table shows the orthogonal regularization is necessary to learn the good trade-off between the energy constraint and the model's learning ability. By visualizing matrices $W$ in EGNN, we observe $W$ will be updated lightly around the initialized matrices to adapt to the downstream node classification tasks. Such weight optimization is important for the attributed graph with plenty of informative node features.
>
> | Datasets | Cora | Pubmed | Coauthors-Physics | Ogbn-arxiv |
> | :---:        |   :----: |    :---: |    :---: |  :---: |
> | # Layer | 64 | 64 | 32 | 32|
> | Constant $W$ | 82.5 | 79.9 | 92.8 | 71.7 |
> | Orthogonal regularization | 85.7 | 80.1 | 93.3 | 72.7 |
>
>
> Q4: Is the energy bounded in Ogbn-arxiv?
>
> Based on the default setting in Section A.4, we show the Dirichlet energies $E(X^{(k)})$ of $32$-layer EGNN here: [1.53, 1.61, 1.57, 1.53, 1.46, 1.42, 1.38, 1.33, 1.29, 1.26, 1.24, 1.23, 1.22, 1.21, 1.21, 1.20, 1.20, 1.19, 1.19, 1.19, 1.18, 1.17, 1.17, 1.16, 1.16, 1.14, 1.11, 1.09, 1.06, 0.99. 0.97, 1.09] * $10^6$. For each layer, as defined in our Dirichlet energy constrained learning principle in Eq. (5), the upper limit is $E(X^{(0)}) = 1.74*10^6$, and the lower limit is $0.75 E(X^{(k-1)})$. It is observed EGNN is still bounded within the pre-defined ranges even with small $\gamma$ in Ogbn-arxiv.
>
> As illustrated in Figure 5 in the supplementary, EGNN is not sensitive to $\gamma$ in Ogbn-arxiv. Once weight $W$ is properly initialized, it is regularized and updated at the vicinity of the sparse diagonal matrix. Even with small $\gamma$, the eigenvalues of updated sparse $W$ will still be determined by the dominant diagonal values, which are close to the carefully initialized ones. Therefore, the Dirichlet energy is still bounded within the pre-defined limits. This also explains why the sparse diagonal weight works better than the other dense orthogonal weight as concerned in Q1, 2.
>
> Q5: Overwhelming residual information.
>
> We use the case of overwhelming residual information to study the importance of residual connection strength in setting up deep GNNs. We build up the relationship between the residual connection and the lower energy limit. As shown in Lemma 4, we prove that the residual connection strength of $c_{\min}$ is actually to ensure the lower limit of Dirichlet energy at each layer in EGNN, i.e., $E(X^{(k)}) \geq c_{\min} E(X^{(k-1)})$. From the Dirichlet energy perspective, the large value of $c_{\min}$ shrinks the available energy range in Eq. (6), and may prevent model from iterating to the optimal point.
>
> The case of overwhelming residual information is worthy of discussing. The ablation study of residual strength is absent in the previous work. For example, JKNet [26] concatenates node embeddings from all the prior layers to predict node labels, where the overwhelming residual information may lead to poor performance as shown in Table 1. Even more, people claims that MLP could achieve the comparable performance as GCN, which is equivalent to the residual strength of 1. We empirically show that the appropriate residual strength can help setting up deep GNNs and delivers the outperforming performance.
>
> Q6:  "GCN" and "Lower" are flipped?
>
> The legends are correct. In GCN, the Dirichlet energies decrease from $0.01$ (at layer $1$) to $0$ (at layer $64$), which are much smaller than those of EGNN or GCNII. The lower limit in EGNN is defined by $c_{\min}E(X^{(k-1)})$, which is obviously larger than zero.
>
> Due to the over-smoothing, the node embdedings in GCN converge to zero vectors and lead to gradient vanishing. Even at the first several layers of GCN, they fail to be updated well to learn the distinguishable node embeddings. Therefore, the Dirichlet energies at all the layers approximate to zero as illustrated in Figure 1.
>
>
> Q7: Can you comment on $\gamma$?
>
> First, as show in Figure 2 in the manuscript and Figures 3, 4 \& 5 in the supplementary, we would like to point out EGNN is extremely robust to the choice of $\gamma$. It is shown that EGNN approximates the optimal performance once $\gamma$ is larger than a specific threshold. The thresholds are $0.3$ in Cora, $0.1$ in Pubmed and Coauthor-Physics, and 1e-4 in Ogbn-arxiv, respectively. The threshold depends on the specific dataset: while a larger $\gamma$ potentially works in the small dataset to strictly regularize Dirichlet energy, a smaller one would be preferred for the large dataset to capture the complex data manifold.
>
> For the small datastes, we use $\gamma=20$ just to emphasize the importance of orthogonal weight controlling. The comparable promising performance could also be achieved with $\gamma$ slightly larger than the threshold. For the large dataset Ogbn-arxiv, $\gamma=1\mathrm{e-}4$ is used since more freedom is required for the model to adapt to  the complex data. As answered in Q4, the energy is still bounded due to the sparse diagonal initialization and regularization to control the dominant eigenvalues.
>
> Q8: GCNII.
>
> Comparing with GCNII, the main differences regarding to the residual connection are: (1) The residual connections to both $X^{(0)}$ and $X^{(k-1)}$ are used, which is more general to unify the existing deep GNNs as analyzed in Section 4.4; (2) We build up the inherent theoretical connection between the residual connection and the lower energy limit in Lemma 4.
>
> Although GCNII is bounded within the energy limits, most of the previous works (including GCNII) are designed based on the empirical experiences motivated from CV domain. The existence of diverse tricks makes the future development a difficult job: How can we combine the different tricks or improve the specific module to formulate the promising GNN model? Without the theoretical insights on pointing out how to build and train deep GNNs, the future development will be limited in the endless and laborious
> neural architecture tuning. In this paper, we make the following two significant theoretical contributions: (1) We point out the important role of Dirichlet energy, and theoretically derive the constrained learning principle towards deep GNNs. (2) We optimize EGNN design from three architecture perspectives with both the theoretical guarantees and the empirical effectiveness, i.e., orthogonal weight, residual connection, and SReLU. As discussed in Section 4.4, our principle and model can well explain the previous works. Based upon our unified framework with solid theory support and empirical engineering, we believe more promising models could be developed in the future, such as the advanced orthogonal weight initialization. Therefore, we still need to take care of Dirichlet energy given its theoretical potential to analyze and tackle the over-smoothing issue.
>
> Q9: semi-supervised?
>
> Yes, it is the semi-supervised node classification setting. The detailed train/valid/test division is given in Section A.1 in the supplementary, which follows the common setting in literature.

---

> > ### Author Response · Authors · 2021-08-31
> > **Further questions**
> >
> > Dear Reviewer:
> >
> >   Thanks for your initial comments. As the discussion period is approaching its end, we would really appreciate it if you could kindly let us know there are any further questions. We will be more than happy to address them fully.
> >
> > Best,
> > Authors

---

### Official Review · Reviewer_tjhZ · 2021-07-16

**Rating:** 6
**Confidence:** 3

**Summary:**

In order to solve the over-smoothing issue for graph neural network training, this paper propose a way with lower and upper constraints in terms of Dirichlet energy at each layer.  According to their experiment results, they achieve state-of-the-art performance by using deep layers in this Energetic Graph Neural Networks (EGNN) framework.

**Limitations And Societal Impact:**

Limitations from previous reviewers are almost solved.

**Main Review:**

Originality:

In this work,  lower and upper constraints in terms of Dirichlet energy at each layer are used for solving over-smoothing issue. This is a new and interesting approach.

Quality:

 Claims are reasonably supported. The initializations satisfy the lower and upper constraints. However, learning with strong regularizations respect to initializations does not mean that the lower and upper constraints hold after the parameters are updated.

Clarity:

The writing could be improved. For example, the abbreviation "EGNN" appear in the abstract, however, it does not state that it means "Energetic Graph Neural Networks".

Significance:

The results seems promising. However, learning with strong constraints respect to the distance with initializations  could limit the ability of graph neural network. Some discussions could be done.



**Time Spent Reviewing:**

5

---

> ### Author Response · Authors · 2021-08-10
> **Response to Reviewer tjhZ**
>
> We really thank the reviewer for the recognition of the originality of our Dirichlet energy constrained learning principle and the theoretically sounded design of EGNN. We would like to provide more explanations to address the reviewer's concerns.
>
> Q: Learning with strong regularizations respect to initializations does not mean that the lower and upper constraints hold after the parameters are updated.
>
> As mentioned by the reviewer, even with the orthogonal weight initialization, we cannot theoretically ensure the lower and upper constraints hold during the training of EGNN. Instead, we adopt the orthogonal regularization due to the following two reasons. First, the direct optimization with the lower and upper energy bounds in Eq. (6) is extremely time-consuming. The weight penalty in Eq. (7) is convex, and could be efficiently optimized. Second, considering the sparse diagonal initialization, the regularization in Eq. (7) updates weight $W$ at the vicinity of initialized matrix. The eigenvalues of updated sparse $W$ will be mainly determined by the dominant diagonal values, which are close to the initialized ones. Thus, the Dirichlet energy at EGNN is still potential to be well bounded, which is empirically validated by Figure 1.
>
> Q: Abbreviation of EGNN in abstract.
>
> Thanks for the careful reviews. We will carefully fix the problem in the revised version.
>
> Q: Learning with strong constraints respect to the distance with initializations could limit the ability of graph neural network.
>
> As show in Figure 2 in the manuscript and Figures 3, 4 \& 5 in the supplementary, we have to point out EGNN is robust to the choice of $\gamma$. It is observed that EGNN approximates the optimal performance once $\gamma$ is larger than a specific threshold. The thresholds are $0.3$ in Cora, $0.1$ in Pubmed and Coauthor-Physics, and $10^{-4}$ in Ogbn-arxiv, respectively. The regularization strength $\gamma$ does not have to be strong to limit the model's learning ability. However, a loose constraint with $\gamma=0$ will damage the performances on Cora, Pubmed, and Coauthor-Physics.

---

> > ### Comment · Reviewer_tjhZ · 2021-09-11
> > **Thanks for your response!**
> >
> > Thank you so much for your clear response. It is more clear about the effect of the regularization and weight initialization.

---

### Official Review · Reviewer_vMYT · 2021-07-16

**Rating:** 7
**Confidence:** 4

**Summary:**

This paper proposes Dirichlet energy constrained learning as a generalizable principle to guide the training of deep GNNs through regularizing Dirichlet energy at each layer. Following this principle, a novel deep architecture called Energetic Graph Neural Networks (EGNN) is proposed, which could learn an optimal Dirichlet energy to enable effective training of deep networks. Empirical results demonstrate that EGNN can be trained to achieve better results with more layers and outperform other GNN deepening approaches.

**Limitations And Societal Impact:**

No.

**Main Review:**

This paper leverages a metric of Dirichlet energy of node embeddings to provide a better understanding of the over-smoothing and over-separation issues of deep GNNs. The new Dirichlet energy constrained learning framework is well presented to explain the key constraints in training deep GNNs. Three learning strategies (orthogonal weight controlling, lower-bounded residual connection, and shifted ReLU activation) are proposed to ensure that the training of the proposed EGNN model satisfies the constrained learning. Theoretical proofs and empirical results on node classification both demonstrate the effectiveness of the proposed solution. This paper is well-written and has made solid technical contributions towards effectively training deep GNNs. Its main claims are theoretically and experimentally supported.

I have carefully read the authors' responses -- Although there are a few potential issues raised by other reviewers, I think overall this is a pretty good paper in my batch, so I retain my recommendation.

**Time Spent Reviewing:**

3 hours

---

> ### Author Response · Authors · 2021-08-10
> **Response to Reviewer vMYT**
>
> We really appreciate the reviewer for the agreements on our theoretical grounding, empirical effectiveness, technical novelty, and expression clarity. Besides the significant technical contributions, as discussed in Section 4.4, our Dirichlet energy constrained learning principle and EGNN model could well explain and connect to the existing works of deep GNNs. Based upon our unified framework with solid theory support and empirical engineering, we believe more the promising models could be developed in the future. For example, the advanced orthogonal weight initialization and regularization can be designed to learn the optimal Dirichlet energy.

---

### Official Review · Reviewer_DaEi · 2021-07-16

**Rating:** 6
**Confidence:** 4

**Summary:**

This paper proposes a Dirichlet energy-based method to alleviate the over-smoothing and over-separating problems which are commonly observed in deep GNNs. Different from existing methods that rely on techniques from CNNs or heuristic strategies, this work provides theoretical proof of the lower and upper bound of the Dirichlet energy in GNNs and proposes EGNN which enables deeper GNNs. The effectiveness of EGNN is evaluated on commonly used benchmarks.

**Limitations And Societal Impact:**

see above

**Main Review:**

Pros:
1.	The paper points out that the Dirichlet energy of each layer of GNN should not too large or too small.
2.	According to the theoretical findings, the paper proposes constraints on Dirichlet energy. Especially, it smartly binds the lower bound of the Dirichlet energy to avoid over-smoothing.
3.	The experimental results show the great impact of Dirichlet energy constraints for deep GNNs.

Cons:
1.	In deep graph learning, solving the Dirichlet Energy of embedding matrix is equivalent to a set of linear equations in terms of Laplacian matrix and embedding matrix. Therefore, the theoretical findings of Lemma 2 and Lemma 3 are equivalent to the theoretical findings of Theorem 3 in [1]. In [1], it also proves the upper bound of $s_max$ in terms of node size. Therefore, the theoretical contribution is not significant.
2.	[2] also constrains the Dirichlet energy by minimizing it although it does not constrain the lower bound of the Dirichlet energy (see Eq. 5 in [2]). However, the paper does not discuss with [2].
3.	The performance improvement is insignificant compared with GCNII [27]. Furthermore, the classification accuracy of GCNII shown in Table 1 is not consistent with the results reported in the original paper [27]. For example, GCNII achieves 80.2 on Pubmed with num_layer=16 [27], while Table 1 reports 79.8.

Questions:
1.	What is your main Theorem? There are only Lemmas in the paper.
2.	In Figure 1, why the Dirichlet energy of GCN is zero at the very beginning? It is supposed to be decreasing to zero after several layers.

[1] Oono, Kenta, and Taiji Suzuki. "Graph Neural Networks Exponentially Lose Expressive Power for Node Classification." International Conference on Learning Representations. 2019.
[2] Deng, C., et al. "GraphZoom: A Multi-level Spectral Approach for Accurate and Scalable Graph Embedding." The International Conference on Learning Representations (ICLR). 2020.


**Time Spent Reviewing:**

6 HOURS

---

> ### Author Response · Authors · 2021-08-10
> **Response to Reviewer DaEi**
>
> Thanks for accepting the technical contributions of Dirichlet energy constrained learning principle and the empirical effectiveness of EGNN. We present our responses below.
>
> Cons 1: Comparison to prior work [1].
>
> We do not agree with this comment that our theoretical contribution is not significant comparing with [1]. We would like to clarify the misunderstanding of prior work [1], and re-claim our contributions as follows.
>
> Based on the Laplacian adjacency matrix and embedding matrix, most of the existing spatial GNNs are implemented by following the recursive neighbor aggregation and feature transformation mechanisms. There already has been several works in literature to explain the over-smoothing issue. While reference [1] relates GNNs with dynamic system and proves node embeddings converge to the invariant space with the increasing of layers, paper [32] cited in our paper uses a more expressive and simple metric (i.e., Dirichlet energy) to recover the findings in [1]. The theoretical result in terms of the upper bound only depends on the eigenvalues of adjacency matrix and the singular values of embedding matrix (e.g., Theorem 2 in [1] and Theorem 3.4 in [32]). The upper bound related to node number in [1] is only for the special case of Erdos–Renyi graph, instead of the general realistic graphs. However, both of them are limited in the theoretical explanation of over-smoothing phenomenon, being short of specific improvement to build up deep GNNs.
>
>
> In this paper, our theoretical contributions are not limited to the upper bound analysis as in [1, 32]. Besides the proposition of constrained learning principle in Proposition 1, the key modules in our EGNN are proved to satisfy the lower and upper energy limits in Lemmas 3, 4, 5. To be specific, we adopt the concept of Dirichlet energy from [1], and overcome the over-smoothing issue theoretically and empirically with the following contributions: (i) Based on the upper bound analysis in [1], we further derive the lower bound of Dirichlet energy in Lemma 1, and relax the energy bounds in Lemma 2 to facilitate the following developments. More importantly, we argue that the Dirichlet energy of node embeddings should be constrained to avoid over-smoothing and over-separating. In Proposition 1, we thus propose Dirichlet energy constrained learning principle to regularize the lower and upper limits of Dirichlet energy towards training deep GNNs successfully. (ii) According to the principle, we optimize the design of deep GNNs empirically from three perspectives, including the orthogonal weight controlling, lower-bounded residual connection, and SReLU activation. Each of them is accompanied with theoretical analysis to demonstrate its rationality -- the Dirichlet energy is constrained well as defined in the proposed principle. Specifically, we prove in Lemma 3 to demonstrate that the orthogonal weight initialization could satisfy the upper limit of constrained learning principle. We prove in Lemmas 4 and 5 to show that the residual connection could constrain Dirichlet energy within the appropriate limits. We cite Lemma 6 to motivate us to incorporate SReLU in our model.
>
> In summary, comparing with [1] from the theoretical perspective, we propose the constrained learning principle in Proposition 1, and prove how the key modules in EGNN are designed to satisfy the principle in Lemmas 3, 4, 5. In addition, comparing with [1] from the engineering perspective, we deliver EGNN model to achieve better performance with more layers, which means the over-smoothing issue is successfully tackled. Comparing with the previous works focusing on either pure analysis or heuristic engineering, we unify both the theory supports and the empirical architecture design to guide the development of deep GNNs. Based upon our unified framework, we believe more the promising models could be developed in the future.
>
> Cons 2: Discussion of [2].
>
> Dirichlet energy is a common metric to quantify the graph smoothness in literature. It is usually required to be  small to make connected nodes close in the embedding space. Paper [2] optimizes Dirichlet energy of Eq. (5) with the approximated iterative solution as shown in Eq. (7), which is a GNN variant parameterized with iteration time $k$. According to the public repository of [2], the authors use a default value of $k=1$ to reduce Dirichlet energy roughly for the coarsened graph, since their main goal is how to coarsen the large-scale graph. In this paper, we target at tackling the over-smoothing issue brought by a large value of $k$. The node embeddings will converge to meaningless unique equilibrium when $k\rightarrow \infty$, which also applies to paper [2]. We point out the importance of constraining the lower and upper limits of Dirichlet energy during the training of deep GNNs. In addition, the approaches to optimize Dirichlet energy are different. While paper [2] directly averages node embeddings as shown in Eq. (7), we exploit three architecture modules (i.e., orthogonal weight, residual connection, and SReLU) to constrain Dirichlet energy.
>
> In summary, comparing with reference [2], we are different in the main goal and the optimization solution of Dirichlet energy. Therefore, reference [2] is not necessary to be discussed in the our work targeting at solving over-smoothing issue.
>
> Cons 3: Performance of GCNII.
>
> For the accuracy improvement, by reviewing the recent deep GNNs [I], the absolute improvement larger than 0.2% is widely accepted to be significant enough in the semi-supervised transductive learning setting, especially for the benchmarks Cora and Pubmed. Considering the official leaderbord of Ogbn-arxiv [II], comparing with GCNII, the absolute improvement of 2.2% in the $32$-layer case is even more convincing. Since the open code of GCNII is evaluated with only one random seed [III], we test GCNII by directly copying its implementation to our environment with $10$ random initializations. All the experiment results are recorded by the average accuracy. For the deep cases (e.g., $64$ layers in Cora and Pubmed), the reported baseline performances are close or even better than those in GCNII paper [27].
>
> Besides the empirical effectiveness, we would like to emphasize our theoretical contributions: (1) The Dirichlet energy constrained learning principle states how to constrain Dirichlet energy towards successfully training deep GNNs; (2) The three architectural modules in EGNN are designed with comprehensive theoretical proofs to satisfy the upper and/or lower limits as defined in the principle. Instead, GCNII is designed purely based on the empirical experience in graph representation learning. Our unified framework with sounded proofs and empirical effectiveness deliver significant technical contributions comparing with GCNII and the other heuristic methods.
>
> [I] https://github.com/mengliu1998/awesome-deep-gnn
>
> [II] https://ogb.stanford.edu/docs/leader_nodeprop/#ogbn-arxiv
>
> [III] https://github.com/chennnM/GCNII/blob/master/PyG/cora/cora.py
>
> Q1: What is your main Theorem?
>
> Our main theorems are: (1) We extend to analyze the lower bound of Dirichlet energy in Lemma 1, and relax the bounds in Lemma 2 to facilitate the following design of guiding principle and EGNN. More importantly, we propose the Dirichlet energy constrained learning principle in Proposition 1 to point out the crucial role of  Dirichlet energy in training deep GNNs. (2) For the architectural design of EGNN, we prove in Lemma 3 to show that the orthogonal weight initialization is guaranteed to satisfy the upper energy limit. In Lemmas 4 and 5, we prove EGNN equipped with residual connection will be constrained within the lower and upper energy limits.
>
> We have to reiterate our two main contributions: (1) We theoretically derive the constrained learning principle to regularize Dirichlet energy towards deep GNNs. (2) We optimize EGNN from three architecture perspectives with both the theoretical guarantees and the empirical effectiveness, i.e., orthogonal weight, residual connection, and SReLU. As discussed in Section 4.4, our principle and EGNN can explain the existing works. Therefore, our theorems about the constrained learning principle (in Proposition 1) and the architecture designs (in Lemmas 3, 4, 5) would be beneficial to guide the future developments. For example, the other orthogonal initialization and residual connection pattern could be proposed by following our theoretical findings.
>
> Q2: Figure 1.
>
> In GCN, the Dirichlet energies shrink from $0.01$ (at layer $1$) to $0$ (at layer $64$), which are much smaller than those of EGNN or GCNII. Due to the over-smoothing, the node embdedings in GCN converge to zero vectors and lead to gradient vanishing. Even at the first several layers of GCN, they fail to be updated well to learn the distinguishable node embeddings. Therefore, the Dirichlet energies at all the layers approximate to zero as illustrated in Figure 1.

---

> > ### Author Response · Authors · 2021-08-31
> > **Further questions**
> >
> > Dear Reviewer:
> >
> >   Thanks for your initial comments. As the discussion period is approaching its end, we would really appreciate it if you could kindly let us know there are any further questions. We will be more than happy to address them fully.
> >
> > Best,
> > Authors

---

> > ### Author Response · Authors · 2021-09-02
> > **Appreciate further comments**
> >
> > Dear Reviewer,
> >
> >   We really appreciate your previous constructive comments, and have addressed your concerns one by one. Since the discussion period is approaching its end, please kindly let us know if you have further comments.
> >
> > Best,
> >
> > Authors

---

> > ### Comment · Reviewer_DaEi · 2021-09-02
> > **Reply**
> >
> > Thank you very much for your response. I have read the feedback and the other reviewers' comments. The authors addressed some of my questions. I would like to raise my rating to 6. However, there are still some question remaining unclear.
> >
> > Q1: I think the authors may misunderstand my comments. I was not asking for the main contribution of the Theorem. I was arguing the writing issue of the paper. Since the paper provides some theoretical results, a main theorem as “Theorem 1: XXX” should be appeared in the paper. However, only Lemma 1 throughout Lemma 6 are listed in the paper. So what are these Lemmas supporting for? Should you make some Lemma as Theorem and some Lemma as Remarks to make the logic clear.
> >
> > Cons 3: I have a similar concern with Reviewer TzoX’s 8th comment: the results are so close to GCNII. Since the authors claim that they run the experiments with 10 random initializations, please report both mean and std of the performances here.
> >
> > Also, I agree the point 3 of Reviewer TzoX. When W’s are close to the identity, it makes the corresponding layer close to $((1-\beta)I + \beta W)$ portion in GCNII. That is why your results is very close to the results in Table2 of GCNII. BTW, I should point out an error in the response to Reviewer TzoX. SGC does not set all weights W to be identity. It removes the non-linearity, ReLU.
> >
> > Minors:
> > Cons 1: I agree that Lemma 4-5 are the new thermotical results. But for Lemma 2, it is an incremental extension in terms of the Theorem 3 in [1]. Please add the corresponding citation ([16] in the paper) for Lemma 2.
> >
> > Cons 2: GraphZoom also leverages the Dirichlet Energy in their work. I think this is a related work. Please discuss it in the revised version

---

> > > ### Author Response · Authors · 2021-09-04
> > > **Thanks for your feedbacks**
> > >
> > > We sincerely thank for your constructive feedbacks, and would like to address the comments to make them clear.
> > >
> > > Q1 and Cons1: Distinguish Lemma and Theorem.
> > >
> > > You are correct in understanding our Lemmas and Theorems. Lemmas 1 and 2 are the extensions from references [1, 32], Lemmas 3, 4, and 5 are our main theorems. As shown in Line 105 in the manuscript, we have cited related work [32] to claim that we apply the existing upper bound analysis, and further derive the lower bound in this paper. In the revised version, we will correctly cite [1, 32] in Lemmas 1 and 2, and rename Theorems 3, 4, and 5 to make the logic more clear.
> > >
> > > Cons3: Results.
> > >
> > > Due to the space limit, we ignored the standard variances in the manuscript. To address the concern about the results, we list both the mean and standard variance in the following table, where the percentile precision of the accuracy in percent is used to make the comparison more correct.
> > >
> > > According to the table, we have the close performances with GCNII only on dataset Pubmed, where our EGNN consistently delivers absolute improvements larger than 0.16%. For the semi-supervised learning on datasets Cora and Coauthor-Physics, we would like to emphasize the absolute improvements around 0.3\% are significant enough in the deep cases (32 or 64), and we also have smaller variances comparing with GCNII. For the cases of fewer layers (2 or 16) on Cora, our improvements over GCNII are larger than 0.7%. Furthermore, on the larger dataset Obgn-arxiv, the absolute improvement of 2.2% in 32 layers is even more convincing by reviewing the leaderboard. Therefore, our model does deliver significantly better performances than GCNII.
> > >
> > > Comparing with $(1-\beta)I+\beta W$ adopted in GCNII, we instead use the orthogonal initialization and regularization to let model learn the trade-off between orthogonality and learning ability depending on the downstream tasks. We theoretically show the benefit of applying orthogonal weights, which aim to control their eigenvalues to bound the Dirichlet energy. In contrast, GCNII explicitly uses $\beta$ to get the trade-off, and applies the identity matrices purely based on empirical experiences.
> > >
> > > | Datasets      | Cora | Cora | Cora | Pubmed | Pubmed | Pubmed | Coauthors-Physics | Coauthors-Physics | Coauthors-Physics | Ogbn-arxiv | Ogbn-arxiv | Ogbn-arxiv |
> > > | :---:   |  :----: | :----: | :----: | :---: | :---: | :---: | :---: | :---: | :---: | :---: | :---: | :---: |
> > > | Layer Num | 2 | 16 | 64 | 2 | 16 | 64 | 2 | 16 | 32 | 2 | 16 | 32 |
> > > |GCN| 82.52±0.45 | 22.02±6.24 | 21.86±8.04 | **79.66±0.29** | 37.94±0.53 | 38.42±1.01 | 92.36±0.14 | 13.55±2.88 | 13.13±2.80| 70.42±0.21 | 70.64±0.25 | 68.51±0.96 |
> > > | PairNorm | 74.46±3.13 | 44.23±7.26 | 14.22±1.93| 73.84±0.90  | 68.59±7.30 | 60.03±10.23 | 86.32±1.06 | 84.02±3.34 | 83.56±5.08 | 67.56±0.14 | 70.37±0.15 | 69.63±0.21 |
> > > | DropEdge | 82.73±0.60 | 23.64±7.61 | 25.18±9.20 | 79.55±0.50 | 45.92±7.14  | 39.97±1.14 | 92.46±0.14 | 85.10±3.02 | 35.23±17.5 | 70.49±0.23 | 70.41±0.55 | 67.14±1.78 |
> > > | SGC | 75.68±0.04 | 72.10±0.00 | 24.10±0.00 | 76.05±0.05 | 70.20±0.00 | 38.19±0.03 | 92.19±0.00 | 91.74±0.00| 84.80±0.01 | 69.19±0.04 | 64.01±0.05 | 59.46±0.07 |
> > > | JKNet | 80.84±0.66 | 74.54±3.72 | 70.01±7.66 | 77.15±0.68 | 69.98±6.26 | 66.16±8.35 | **92.71±0.19** | 92.15±0.49 | 91.65±1.35| **70.62±0.11** | 71.85±0.15 | 71.44±0.35 |
> > > | APPNP | 82.94±0.56 | 79.38±0.62 | 79.49±0.66 | 79.33±0.48 | 77.07±0.66 | 76.83±0.90 | 92.33±0.15 | 92.65±0.46 |92.63±0.42 | 68.26±0.78 | 65.47±0.23 | 60.71±0.16 |
> > > | GCNII | 82.42±0.45 | 84.55±0.43 | 85.44±0.58 | 77.49±1.91| 79.83±0.56 | 79.94±0.33| 92.49±0.36 | 92.87±0.23 | 92.94±0.15 | 70.09±0.27 | 71.46±0.16 | 70.52±0.30|
> > > | EGNN | **83.18±0.24** | **85.36±0.35** | **85.71±0.55** | 79.17±0.34 | **79.99±0.36** | **80.10±0.26** | 92.59±0.09 | **93.10±0.16** | **93.31±0.12** | 68.41±0.25 | **72.74±0.23** | **72.74±0.35** |
> > >
> > > Cons3: SGC.
> > >
> > > Thanks for your correction. Actually, the graph convolution at the k-th hidden layer of SGC could be expressed as: $X^{(k)} = \tilde{P}X^{(k-1)} = \tilde{P}X^{(k-1)}I_d$. That means SGC removes both the non-linear activation and trainable weights in the hidden layers. For the part of removing trainable weights, it could be regarded as setting all the weights to be identity matrices in the hidden layers.
> > >
> > > Cons2: GraphZoom.
> > >
> > > Thanks for your careful review. We will include GraphZoom and more other related work of Dirichlet energy in the revised version.

---

### Author Response · Authors · 2021-08-27
**Appreciating for your previous efforts and further comments**

Dear AC and Reviewers,

We genuinely thank AC and all the reviewers for your time and the constructive comments! Hope our previous responses have addressed your concerns.

Specifically, we clarify the misunderstandings of prior works and our main theorems raised by Reviewer DaEi, and provide more experiments to address the concerns of orthogonality raised by Reviewer TzoX. Comparing with the previous heuristic methods, we would like to  reiterate our three main contributions: (1) We exploit Dirichlet energy to analyze the over-smoothing & over-separating, and propose the constrained learning principle towards training deep GNNs. (2) We optimize GNN design from three architectural perspectives with both theoretical guarantees and empirical effectiveness, i.e., orthogonal weight, residual connection, and SReLU. (3) We make the initial step to provide an unified framework with theoretical supports and empirical engineering, and can explain most of the existing works. Based upon our framework, more the promising techniques could be developed to boost deep GNNs in the future, such as the orthogonal weight initialization and regularization.

As the discussion period is approaching its end, we would really appreciate it if you could kindly let us know there are any further questions. We will be more than happy to address them fully.

Yours Sincerely,
Authors

---

### Decision · Program_Chairs · 2021-09-27

**Decision:**

Accept (Poster)

**Comment:**

Congratulations! Your paper is accepted to NeurIPS 2021.
Please incorporate the edits and corrections as discussed in the rebuttal.